# Function Classes for Identifiable Nonlinear Independent Component Analysis

**Simon Buchholz**[1]          **Michel Besserve**[1]          **Bernhard Schölkopf**[1]

[1]Max Planck Institute for Intelligent Systems, Tübingen, Germany

## Abstract

Identifiability issues constitute a major obstacle for *Unsupervised Causal Representation Learning*, which aims at learning the ground-truth data generating mechanisms from unlabelled observations. Insights about this problem are provided by the field of *Independent Component Analysis*, studying models that maps statistically independent latent variables to observations via a deterministic, possibly nonlinear function. Indeed, several families of *spurious solutions* fitting perfectly the data, but that *do not* correspond to the ground truth factors, can be constructed in generic nonlinear settings. However, recent work suggests that constraining the function class of such models may promote identifiability. Specifically, function classes with constraints on their partial derivatives, gathered in the Jacobian matrix, have been proposed, such as orthogonal coordinate transformations (OCT), which impose orthogonality of the Jacobian columns. In the present work, we prove that a subclass of these transformations, conformal maps, is identifiable and provide novel theoretical results suggesting that OCTs have properties that prevent families of spurious solutions to spoil identifiability in a generic setting.

## 1 INTRODUCTION

Unsupervised representation learning methods can fit Latent Variables Models (LVM) to complex real world data. While those latent representations allow to create realistic novel samples or represent the data in a compact way Kingma and Welling [2014], Goodfellow et al. [2014], they are a priori not related to the underlying ground truth generative factors of the data. Recently, the desiderata has emerged to learn *causal representations* that reflects the true underlying data-generating factors and mechanisms, as they are expected to help with various downstream tasks, e.g., out of distribution generalisation Schölkopf et al. [2021], Bengio et al. [2013].

One principled framework for representation learning is Independent Component Analysis (ICA) where one tries to recover unobserved sources $s \in \mathbb{R}^d$ from observations $x = f(s)$ and one assumes that the components $s_i$ are independent. An important result is that for linear functions $f$ it is possible to recover $s$ from observations $x$ up to certain symmetries, i.e., the model is *identifiable* Comon [1994]. In contrast, for general non-linear models $f$ is highly non-identifiable Hyvärinen and Pajunen [1999]. This has important consequences for representation learning, in particular the learning of disentangled representations is also unidentifiable without some access to the underlying sources Locatello et al. [2019]. Notably, this makes theoretical analysis of the large body of methods (see, e.g., Higgins et al. [2017], Kim and Mnih [2018], Ridgeway and Mozer [2018]) that enforces disentanglement difficult.

Several additional assumptions were suggested to make the ICA problem identifiable. Broadly, there are two directions. First, some works imposed additional or different restrictions on the distribution of the sources. One line of research adds temporal structure by considering time series data Harmeling et al. [2003], Hyvärinen and Morioka [2016, 2017]. More recently, Hyvärinen et al. [2019] proposed to introduce an observed auxiliary variable $u$, e.g., a class label, such that the source distribution has independent components conditional on the auxiliary variable. They show that under suitable assumptions on the distribution of $u$ and $s$ arbitrary nonlinear mixing function can be identified. Several recent works extended this approach Khemakhem et al. [2020], Sorrenson et al. [2020], Yang et al. [2022].

Another possibility is to restrict the class of admissible functions by considering more flexible classes than just linear functions but not allowing arbitrary non-linear functions. The general aim of this approach is to find sufficiently small function classes such that ICA is identifiable in this class

*Accepted for the Causal Representation Learning workshop at the 38[th] Conference on Uncertainty in Artificial Intelligence* (UAI CRL 2022).

while making them as large as possible to allow flexible representation of complex data and being applicable to real world problems. So far results in this direction are rather limited. It was shown that the post-nonlinear model is identifiable Taleb and Jutten [1999]. Moreover, it has been shown that ICA with conformal maps in dimension 2 is almost identifiable Hyvärinen and Pajunen [1999] and that volume preserving transformations are identifiable (in the auxiliary variable case, combining the two possible restrictions) Yang et al. [2022]. The recent work Zheng et al. [2022] also considers identifiability of nonlinear-ICA, however their definition of identifiability differs from ours.

In this work we extend the previous works by proving new identifiability results for unconditional ICA. Our main focus is conformal maps (i.e., maps that locally preserve angles) and Orthogonal Coordinate Transforms (OCT) (i.e., maps satisfying that $Df^\top Df$ is a diagonal matrix where $Df$ denotes the derivative of $f$). OCTs, that will also call *orthogonal maps* for simplicty, were recently introduced in the context of representation learning in Gresele et al. [2021] where they were motivated using the independence of mechanisms assumption from the causality literature. The main focus of this work is to prove new identifiability and partial identifiability results for this class of functions. Our main contributions are the following.

- We prove that ICA with conformal maps is identifiable in $d \geq 3$.
- We define a new notion of local identifiability (Definition 5) and prove that ICA with orthogonal maps is locally identifiable (Theorem 3). On the contrary we show that ICA with volume preserving maps is not identifiable not even in the local sense (Theorem 5).
- We introduce new tools to the ICA field: our results are based on connections to rigidity theory, restricting the global structure of functions based on local restrictions. Moreover, in contrast to most earlier results that argue locally using results from linear algebra we exploit the global structure of partial differential equations related to the identifiability problem.

## 2 SETTING

Independent component analysis deals with the problem of identifying underlying sources when observing a mixture of the sources. We will consider the following general setting: there exists some random hidden vector of sources $s \in \mathbb{R}^d$ and the observed data is generated by

$$x = f(s), \qquad p_s(s) = \prod_{i=1}^{d} p_i(s_i) \qquad (1)$$

where $f : \mathbb{R}^d \to \mathbb{R}^d$ is a smooth invertible function. The condition on $s$ means that its coordinates (often referred

to as factors of variation) are independent. Formally this means that the distribution of $s$ which we will denote by $\mathbb{P}$ satisfies $\mathbb{P} \in \mathcal{M}_1(\mathbb{R})^{\otimes n}$ where $\mathcal{M}_1(\mathbb{R})$ denotes the probability measures on $\mathbb{R}$. The goal of ICA is to find an unmixing function $g : \mathbb{R}^d \to \mathbb{R}^d$ such that $g(x)$ has independent components. Ideally, this should recover the true underlying factors of variation and achieve *Blind Source Separation* (BSS), i.e., $g = f^{-1}$ up to certain symmetries. Identification of the true generative factors of variations of an observed data distribution is of interest also since these provide a causal and interventional understanding of the data.

An important observation was that in the generality stated above identification of $s$ is not possible. In Hyvärinen and Pajunen [1999] two general constructions of spurious solutions were given, the well known Darmois construction and a construction based on measure preserving transformations. The latter one is closer to our work here and we will discuss those in more detail in Section 4 and Appendix B. In a nutshell it is based on the observation that for measures $\mathbb{P}$ with smooth density one can construct smooth *Measure Preserving Transformations* (MPT), $m : \mathbb{R}^d \to \mathbb{R}^d$ (that mix the different coordinates), i.e., maps that leave $\mathbb{P}$ invariant, such that $m(s) \overset{\mathcal{D}}{=} s$ if $s \sim \mathbb{P}$.[1] This implies that all functions $(f \circ m)^{-1}$ recover independent sources since $(f \circ m)^{-1}(x) \overset{\mathcal{D}}{=} s$ making BSS impossible.

Thus it is a natural question whether additional assumptions on the mixing function $f$ or distribution of $s$ allow us to identify $f$. Let us define a framework for identifiability. We assume data is generated according to (1) where $f \in \mathcal{F}$ for some function class of invertible functions which we will always assume to be diffeomorphisms[2] and we assume the source distribution $\mathbb{P}$ satisfies $\mathbb{P} \in \mathcal{P}$ for some set of probability distributions $\mathcal{P} \subset \mathcal{M}_1(\mathbb{R})^{\otimes d}$. Finally, let $\mathcal{S}$ be a group of transformations $g : \mathbb{R}^d \to \mathbb{R}^d$ that encodes the allowed symmetries up to which the sources can be identified as follows.

**Definition 1.** (Identifiability) We say that independent component analysis in $(\mathcal{F}, \mathcal{P})$ is identifiable up to $\mathcal{S}$ if for functions $f, f' \in \mathcal{F}$ and distributions $\mathbb{P}, \mathbb{P}' \in \mathcal{P}$ the relation

$$f(s) \overset{\mathcal{D}}{=} f'(s') \quad \text{where } s \sim \mathbb{P} \text{ and } s' \sim \mathbb{P}' \qquad (2)$$

implies that there is $h \in \mathcal{S}$ such that $h = f'^{-1} \circ f$ on the support of $\mathbb{P}'$.

Note that we require the identity $h = f'^{-1} \circ f$ only to hold on the support of $\mathbb{P}$ because for complex classes $\mathcal{F}$ there is in general no unique extension of $f$ beyond the support of $\mathbb{P}$ and without data the extension cannot be identified.

---

[1] We use the notation $X \overset{\mathcal{D}}{=} Y$ to indicate that the two random variables $X$ and $Y$ follow the same distribution

[2] A diffeomorphism is a differentiable bijective map with differentiable inverse.

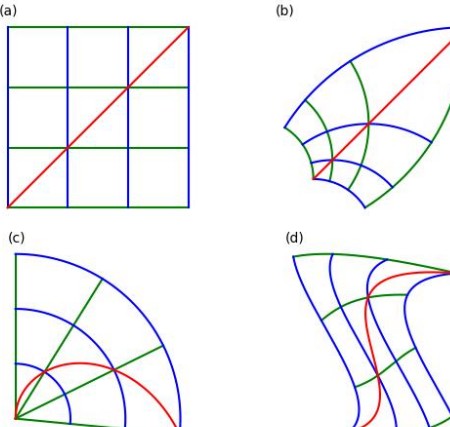

Figure 1: Illustration of the considered function classes. (a) shows a standard coordinate frame, (b) a conformal map applied to this frame which preserves angles, (c) an orthogonal map (polar coordinates) that preserve the orthogonality of lines parallel to the coordinate axes but not all angles (see red line), (d) a volume preserving map.

We do not always make this explicit in the following. Put differently, identifiability means that given observations of $x = f(s)$ and knowledge of $(\mathcal{F}, \mathcal{P})$, we can find $g$ such that $g \circ f \in \mathcal{S}$, in particular the reconstructed sources $s' = g(x)$ and the true sources $s$ are related by a symmetry transformation in $\mathcal{S}$. In Appendix B we will discuss how to identify the set $\mathcal{S}$ and how spurious solutions to the identification problem can be constructed. In the following it will be convenient to use the notation $f_* \mathbb{P}$ which denotes the push-forward of the measure $\mathbb{P}$ along the function $f$. For a formal definition we refer to Appendix A but we note here that the distribution of $f(s)$ equals $f_* \mathbb{P}$ if $s \sim \mathbb{P}$ and (2) can be equivalently written as $f_* \mathbb{P} = f'_* \mathbb{P}'$.

We illustrate Definition 1 through the well known example of linear maps

$$\mathcal{F}_{\mathrm{lin}} = \{f : \mathbb{R}^d \to \mathbb{R}^d : f \text{ is linear and invertible}\}, \quad (3)$$

i.e., $x = As$ for some invertible matrix $A \in \mathbb{R}^{n \times n}$. We further define

$$\mathcal{P}_{\mathrm{lin}} = \{\mathbb{P} \in \mathcal{M}_1(\mathbb{R})^{\otimes d} : \text{at most one } \mathbb{P}_i \text{ Gaussian}\}, \quad (4)$$
$$\mathcal{S}_{\mathrm{lin}} = \{P\Lambda : P \text{ permutation matrix and } \Lambda \text{ diag. matrix}\}. \quad (5)$$

It is easy to check that $\mathcal{S}$ is a group. Then the following identifiability result for $\mathcal{F}_{\mathrm{lin}}$ is well known.

**Theorem 1.** *(Theorem 11 in Comon [1994]) The pair* $(\mathcal{F}_{\mathrm{lin}}, \mathbb{P}_{\mathrm{lin}})$ *is identifiable up to* $\mathcal{S}_{\mathrm{lin}}$.

Moreover, this result is optimal as the ordering and scale of the $s_i$ cannot be identified and the restriction to at most one

Table 1: Overview of new identifiability results. Note that *Identifiable* implies *Locally identifiable* and if *Locally identifiable* does not hold neither of the other two properties can hold.

| $\mathcal{F}$ | Identifiable (Def. 1) | | Locally identifiable (Def. 5) | |
|---|---|---|---|---|
| Linear | ✓ | | ✓ | |
| Conformal | ✓ | (Thm. 2) | ✓ | |
| Orthogonal | ? | | ✓ | (Thm. 3) |
| Volume pres. | ✗ | | ✗ | (Thm. 5) |
| Nonlinear | ✗ | | ✗ | (Lemma 1) |

Gaussian component is required as rotations of Gaussians are Gaussian. For completeness we provide a proof of this result in Appendix C as this serves as a preparation for the more involved Theorem 2 below. In the next sections we discuss our results on identifiability of ICA for different function classes. An illustration of the considered classes can be found in Figure 1.

# 3 RESULTS FOR CONFORMAL MAPS

Our first main result is an extension of Theorem 1 to conformal maps. A conformal map is a map that locally preserves angles, i.e. locally it looks like a scaled rotation. It can be shown that this is equivalent to the following definition.

**Definition 2.** (Conformal map) We define for domains $\Omega \subset \mathbb{R}^d$ the set of conformal maps by $\mathcal{F}_{\mathrm{conf}} = \{f \in C^1(\Omega, \mathbb{R}^d) : Df(x) = \lambda(x)O(x)\}$ where $\lambda : \Omega \to \mathbb{R} \setminus \{0\}$ is a scalar function and $O : \Omega \to \mathrm{O}(d)$ is a map to orthogonal matrices (i.e., $O(x)^{-1} = O(x)^\top$).

For convenience we use the notation $\mathrm{Diag}(d)$ and $\mathrm{Perm}(d)$ for $d \times d$ diagonal and permutation matrices, respectively. We define signed permutation matrices by

$$\mathrm{Perm}_\pm(d) = \{P \in \mathbb{R}^{d \times d} : Q \in \mathrm{Perm}(d), Q_{ij} = |P_{ij}|\}, \quad (6)$$

i.e. the set of matrices whose entry-wise absolute value is a permutation. We define

$$\mathcal{S}_{\mathrm{conf}} = \{x \to \kappa P x + a, \ P \in \mathrm{Perm}_\pm(d), a \in \mathbb{R}^d, \kappa \in \mathbb{R}\} \quad (7)$$

and

$$\mathcal{P}_{\mathrm{conf}} = \mathcal{P}_1^{\otimes n} \cap \mathcal{P}_{\mathrm{lin}}, \qquad \text{where}$$
$$\mathcal{P}_1 = \{\mu \in \mathcal{M}_1(\mathbb{R}), \text{ there is } \emptyset \neq O \subset \mathbb{R} \text{ open,} \quad (8)$$
$$\text{s.t. } \mu \text{ has positive } C^2 \text{ density on } O\}.$$

While this condition might appear a bit technical it actually only rules out pathological cases like the cantor measure or

densities which are nowhere differentiable and probably it could be relaxed further. In particular $\mathcal{P}_1$ contains all probability measures with piecewise smooth densities. Then the following identifiability for conformal maps in dimension $d > 2$ holds.

**Theorem 2.** *For $d > 2$, ICA with respect to the pair $(\mathcal{F}_{\text{conf}}, \mathcal{P}_{\text{conf}})$ is identifiable up to $\mathcal{S}_{\text{conf}}$.*

This means that we can identify conformal maps up to three symmetries, namely constant shifts of the distributions, rescaling of all coordinates by a constant factor, and permutations of the coordinates. The proof is in Appendix D. The main ingredient in the proof is that conformal maps in dimension $d > 2$ are very rigid and can be characterized explicitly as we will discuss in Appendix D.

# 4 RESULTS FOR ORTHOGONAL MAPS

Recently, in Gresele et al. [2021], the more general class of OCTs was considered in the context of ICA. They referred to orthogonal coordinates as IMA maps, referencing to independent mechanisms. This nomenclature was motivated by the causality literature and we refer to their paper for an extensive motivation and further results. As we focus on theoretical results for this function class we stick to the more common term of OCTs. Orthogonal coordinate transformations are defined as the set of functions whose derivative have orthogonal columns, i.e., the vectors $\partial_i f$ and $\partial_j f$ are orthogonal for $i \neq j$.

**Definition 3.** (OCT maps) We define for domains $\Omega \subset \mathbb{R}^d$ the set of OCT maps (orthogonal coordinates) by $\mathcal{F}_{\text{OCT}} = \{f \in C^1(\Omega, \mathbb{R}^d) : Df(x)^\top Df(x) \in \text{Diag}(d)\}$.

First, we note that we can only hope to identify a mechanism $f \in \mathcal{F}_{\text{OCT}}$ up to coordinate-wise transformations and permutations. Therefore we set

$$\mathcal{S}_{\text{OCT}} = \{g : \mathbb{R}^d \to \mathbb{R}^d | g = P \circ h \ \ P \in \text{Perm}_\pm(d),$$
$$h(x) = (h_1(x_1), \dots, h_d(x_d))^\top \text{ with } h_i' > 0\}. \quad (9)$$

It is easy to see that $\mathcal{S}_{\text{OCT}}$ is a group. We note that if $f \in \mathcal{F}_{\text{OCT}}$ and $g \in \mathcal{S}_{\text{OCT}}$ then $f \circ g \in \mathcal{F}_{\text{OCT}}$. Thus, in particular $f_*\mathbb{P} = (f \circ g)_*(g^{-1})_*\mathbb{P}$. This implies that given observations from $f_*\mathbb{P}$ we can identify $f$ and $\mathbb{P}$ only up to $g \in \mathcal{S}_{\text{OCT}}$. More precisely, for any (sufficiently smooth) $\mathbb{P}'$ there is $f'$ such that $f_*\mathbb{P} = f'_*\mathbb{P}'$ where we pick $g$ such that $\mathbb{P}' = g_*^{-1}\mathbb{P}$. [3]

As the distribution of the $s_i$ is not identifiable we will map it to a fixed reference distribution where we will choose the

---

[3]This is possible if both distributions have compact connected support where they have a smooth positive density. We ignore difficulties associated with unbounded support or non-regular measures here

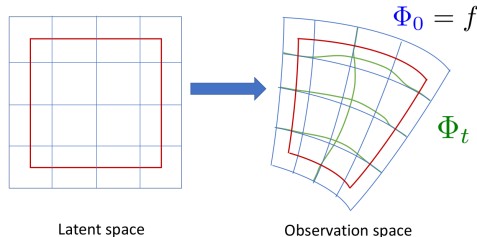

Figure 2: Smooth invariant deformations. The blue grid indicates the transformation $f$, while the green grid shows the deformed map $\Phi_t$. For Definition 5, the red line indicates $\partial\Omega$, outside of the red box $\Phi_t$ is constant.

uniform distribution on $(0,1)^d$. We introduce the shorthand $C_d = (0,1)^d$ for the standard unit cube and denote by $\nu$ the uniform (Lebesgue) measure on $C_d$. For fixed base measure $\nu$ the symmetry group is reduced to permutations and reflections, i.e., maps in $P \in \text{Perm}_\pm(d)$.

We conjecture that for 'typical' pairs of $f \in \mathcal{F}_{\text{OCT}}, \mathbb{P} \in \mathcal{P}_{\text{OCT}}$ ICA is identifiable with respect to $\mathcal{S}_{\text{OCT}}$ (with a suitable definition of $\mathcal{P}_{\text{OCT}}$, e.g., $\mathcal{P}_{\text{OCT}} = \mathcal{P}_{\text{conf}}$). However, we leave a precise statement for future work. We now prove a weaker notion of local identifiability for OCTs.

**Local Stability of OCTs.** We now give partial results towards identifiability of OCTs. While we do not prove general identifiability for OCTs, we demonstrate their *local rigidity*: OCTs cannot be continuously deformed to obtain spurious solutions. This is in stark contrast to the general nonlinear case which we will discuss for comparison below. The result will be based on the following definition.

**Definition 4.** (Smooth invariant deformations) Consider a family of differentiable transformations $\Phi \in C^1((-T,T) \times \mathbb{R}^d, \mathbb{R}^d)$ for some $T > 0$ such that $\Phi_t = \Phi(t, \cdot)$ is diffeomorphism. Let $\mathbb{P}_t \in \mathcal{M}_1(\mathbb{R})^{\otimes d}$ for $t \in (-T,T)$ be a family of probability measures. We call $(\Phi_t, \mathbb{P}_t)$ a *smooth invariant deformation* of the pair $(f, \mathbb{P})$ if $\Phi_0 = f$, $\mathbb{P}_0 = \mathbb{P}$ and $(\Phi_t)_*\mathbb{P}_t = f_*\mathbb{P}$.

An illustration of this definition can be found in Figure 2. Based on invariant deformations we can now define a local identifiability property of ICA in a given function class.

**Definition 5.** (Local identifiability of ICA). Consider a function class $\mathcal{F}$. Let $(\Phi_t, \nu)$ be a smooth invariant deformation of $(f, \nu)$ (i.e., $\mathbb{P}_t = \nu$ is constant). Assume that $\Phi_t \in \mathcal{F}$ and there is an open set $\Omega \subset C_d$ with $\bar{\Omega} \subset C_d$ such that $\Phi_t(x) = f(x)$ for $x \notin \Omega$. Then we say that ICA in $\mathcal{F}$ is *locally identifiable* at $(f, \nu)$ if under these assumptions $\Phi_t = f$ for all $t$.

We call $\mathcal{F}$ *locally identifiable* if it is locally identifiable at $(f, \nu)$ for all $f \in \mathcal{F}$. Local identifiability of a function class means that we cannot smoothly transform a function

$f \in \mathcal{F}$ to any other function $f' \in \mathcal{F}$ such that we stay in $\mathcal{F}$ along the transformation path, the observational distribution remains the same, and we fix the functions close to the boundary of $C_d$. Note that this definition is local in two ways. We consider smooth transformations of the ground truth, i.e., small changes of the data generating function $f$ and in addition we assume that the changes are local in $s$, i.e., sources $s$ close to the boundary are kept invariant. An extension to varying $\mathbb{P}$ is possible but not necessary in the context of OCTs as explained above. We can show that local identifiability holds true in $\mathcal{F}_{\mathrm{OCT}}$.

**Theorem 3.** *The function class $\mathcal{F}_{\mathrm{OCT}}$ is locally identifiable at $(f, \nu)$ for all $f \in \mathcal{F}_{\mathrm{OCT}}$.*

Let us reiterate what this theorem shows (illustrated in Fig. 2): we cannot smoothly and locally transform the function $f$ such that (1) the observational distribution remains invariant, i.e., equal to $f_*\nu$, and (2) the deformed functions remain OCTs. The proof, in Appendix E, introduces new tools to the field of ICA. The main idea is to consider the vector field $X$ that generates the deformation $\Phi_t$ and then rewrite the assumption as systems of partial differential equations for $X$. The proof is then completed by showing that the only solution of this system vanishes.

At a high level this result suggests that OCTs can be identified if we know $f$ close to the boundary of the support of $s$, e.g., by having, in addition to unlabelled data $f(s)$, labelled data $(s, f(s))$ for those $s$ where one coordinate $s_i$ is extremal. Note that we actually do not show this result as there might be further solutions which are not connected by smooth transformations. We expect that this result can be generalised substantially. In particular, we conjecture that for "most" functions $f$ the compact support condition can be removed thus giving a stronger local identifiability result up to the boundary of the support of $s$. As a partial result in this direction we show the following theorem.

**Theorem 4.** *Let $f : C_d \to \mathbb{R}^d$ be given by $f(x) = RDx$, where $R \in \mathrm{O}(d)$ and $D = \mathrm{Diag}(\mu_1, \ldots, \mu_d)$ with $\mu_i > 0$ and $\mu_i^{-2}$ are linearly independent over the rational numbers $\mathbb{Q}$. Suppose that $(\Phi_t, \nu)$ (i.e., $\mathbb{P}_t = \nu$ is constant) is a smooth invariant deformation of $(f, \nu)$ and $\Phi_t$ is analytic in $t$. Then $\Phi_t \in \mathcal{F}_{\mathrm{OCT}}$ for all $t$ implies that $\Phi_t = f$ on $C_d$, i.e., $\Phi_t$ is constant in time.*

The proof is in Appendix E and similar to Theorem 3. Note that for random $\mu_i$ whose distribution have a bounded density the condition is satisfied with probability one. Moreover, we do not expect non-constant $\Phi_t$ to exist when we remove the assumption on the $\mu_i$ but we are unable to show this.

**Comparison with ICA for general nonlinear functions.**
Let us emphasize that those results are non-trivial as they establish a large difference between ICA with generic nonlinear maps and ICA with OCTs. To clarify this we state that

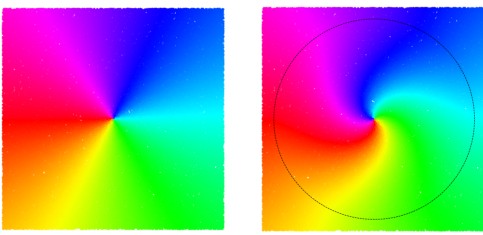

Figure 3: Illustration of radius dependent rotations as defined in Lemma 1. The left figure shows the initial sources. In the right figure a radius dependent volume preserving transformation was applied (see Appendix B).

no result similar to Theorem 3 holds without the assumption that $\Phi_t \in \mathcal{F}_{\mathrm{OCT}}$. Put differently, the function class $\mathcal{F}_{\mathrm{nonlinear}}$ is not locally identifiable.

*Fact* 1. Suppose $f : C_d \to \mathbb{R}^d$ is a diffeomorphism on its image. Then there are many smooth invariant deformations $(\Phi, \mathbb{P}_t)$ of $(f, \nu)$ such that $\mathbb{P}_t = \nu$ and $\Phi_t|_{\Omega^c} = f|_{\Omega^c}$ for any $\Omega \subset C_d$ open.

For completeness we provide a general construction that is close to our proof of Theorem 3 in Appendix B in the supplement. A very clear construction for this result was given in Hyvärinen and Pajunen [1999].

**Lemma 1** (Smoothly varying radius dependent rotations (see Hyvärinen and Pajunen [1999])). *Let $R : \mathbb{R} \times \mathbb{R}_+ \to \mathrm{O}(d)$ be a smooth function mapping to orthogonal matrices and let $a \in C_d$. Assume that $R(t, r) = \mathrm{id}$ for $r \geq \mathrm{dist}(a, \partial C_d)$. Then the map $s \to h_{R,a}(t, s) = R(|s - a|, t)(s - a) + a$ preserves the uniform measure $\nu$ on $C_d$ for all $t$ so that $f \circ h_{R,a}(t, s) \overset{\mathcal{D}}{=} f(s)$ for all $t$ if $s$ is distributed according to $\nu$.*

An illustration of this construction is shown in Figure 3. Clearly, by concatenation this allows us to create a vast family of spurious solutions. Note that those families of solutions are excluded when restricting to OCTs which is a consequence of Theorem 3.

We summarize the results of this section informally (we do not claim that the statements below regarding (infinite dimensional) manifolds can be made rigorous). For a given data generating mechanism $(f_0, \nu)$ we expect that typically the set of all solutions $M_{\mathrm{OCT}} = \{f \in \mathcal{F}_{\mathrm{OCT}} | f_*\nu = (f_0)_*\nu\} \subset \mathcal{F}_{\mathrm{OCT}}$ is a zero dimensional submanifold, i.e., consists of isolated spurious solutions and we prove this when fixing the boundary (see Theorem 3) while the corresponding submanifold of general nonlinear spurious solutions $M_{\mathrm{nonlinear}} = \{f : C_d \to \mathbb{R}^d | f_*\nu = (f_0)_*\nu\}$ is infinite dimensional even when requiring $f(s) = f_0(s)$ close to the boundary of $C_d$.

# 5 RESULTS FOR VOLUME PRESERVING MAPS

Let us finally consider volume preserving transformations. Those are defined as the set of functions

$$\mathcal{F}_{\text{vol}} = \{ f : \Omega \to \mathbb{R}^d \mid \det Df(x) = 1 \text{ for all } x \in \Omega \}. \tag{10}$$

Invertible volume preserving deformations have the property that they preserve the standard (Lebesgue)-measure $\lambda$ in the sense that $f_* \lambda_\Omega = \lambda_{f(\Omega)}$. Recently it was proposed that volume preserving functions are a suitable function class for ICA. Here we show that those functions are not sufficiently rigid to allow identifiability of ICA in the unconditional case. Note that Lemma 1 and Fact 1 already show how to construct spurious solutions for the case that the base distribution is the uniform measure $\nu$. However, for an arbitrary distribution $\mathbb{P}$ this is slightly more difficult because we need to find maps $g$ that preserve $\mathbb{P}$, i.e., $g_* \mathbb{P} = \mathbb{P}$, and are volume preserving, i.e., preserve the standard measure. Nevertheless, we have the following theorem.

**Theorem 5.** *Let $p$ be a twice differentiable probability density with bounded gradient. Suppose that $x = f(s)$ where the distribution $\mathbb{P}$ of $s$ has density $p$ and $f$ is a diffeomorphism with $\det Df(x) = 1$ for $x \in \mathbb{R}^d$. Then there is a family of functions $f_t : \mathbb{R} \times \mathbb{R}^d \to \mathbb{R}^d$ with $f_0 = f$ and $f_t \neq f_0$ for $t \neq 0$ such that $\det Df_t(x) = 1$ and $(f_t)_* \mathbb{P} = f_* \mathbb{P}$.*

The proof and an illustration are in Appendix F.

# 6 DISCUSSION

ICA is long known to be identifiable for linear maps, baring pathological cases, and highly non-identifiable for general nonlinear ones. Surprisingly, similar results for function classes of intermediate complexity remain scarce. In this work we address this question with several identifiability results for different function classes. Our first main result is that ICA is identifiable in the class of conformal maps (up to classical ambiguities). This considerably extends previous claims, limited to a specific 2D setting Hyvärinen and Pajunen [1999], and ruling out several families of spurious solutions Gresele et al. [2021]. On the negative side we show that the ICA problem for volume preserving maps admits a large class of spurious solutions. Finally, we show that OCTs satisfy certain weaker notions of local identifiability.

While the main focus of current research after the seminal work of Hyvärinen et al. [2019] is on the auxiliary variable case, we think that a better understanding of unconditional ICA is still an important challenge. Firstly, it is a fundamental research question that is, as illustrated by our results, deeply rooted in functional analysis. Secondly there is high application potential for completely unsupervised learning without any auxiliary variables, as the corresponding datasets do not require labelling or specific experimental settings. Thirdly, it is very likely that the techniques can be generalised to the auxiliary variable case.

In our proofs, we draw connections to methods and techniques that, to the best of our knowledge, have not been used in the context of ICA before. We relate the identifiability problem in ICA to the rigidity of the considered function class $\mathcal{F}$ and use tools from the theory of partial differential equations. These techniques have been applied very successfully to the analysis of elastic solids Ciarlet [2021, 1997] and we believe that there are many applications of these methods in the field of ICA.

Another important open problem is assessing the type of constraints on ground truth mechanisms, encoded by function classes, that are relevant for real world data. It is plausible that those mechanisms are typically much more regular than generic nonlinear functions. Recently, Gresele et al. [2021] suggested, based on arguments from the causality literature that $\mathcal{F}_{\text{OCT}}$ is a natural class for representation learning (and our results show it also has favourable theoretical properties), but this will require experimental confirmation on real world data. To bridge the gap between theory and real world applications, two questions of considerable interest are (1) an analysis of undercomplete ICA (i.e., $f : \mathbb{R}^d \to \mathbb{R}^{d'}$ with $d < d'$) with restricted function classes which is completely open and (2) identifiability results when we only know that $f$ is close to some function class $\mathcal{F}$.

Finally, a central question from a machine learning perspective is the ability to design learning algorithms that can train LVMs with identifiable function class constraints. Interestingly, Gresele et al. [2021] showed that OCT maps can be learnt using a closed from regularized likelihood loss, thereby providing, supported by our result, a full-fledged identifiable nonlinear ICA framework.

## Acknowledgements

This work was partially supported by the German Federal Ministry of Education and Research (BMBF): Tübingen AI Center, FKZ: 01IS18039B; and by the Machine Learning Cluster of Excellence, EXC number 2064/1 - Project number 390727645.

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

# Function Classes for Identifiable Nonlinear Independent Component Analysis

## Supplementary Material

In the supplement we provide the proofs of our results and we discuss the relevant background.

## A    MATHEMATICAL BACKGROUND

For the convenience of the reader we collect some mathematical definitions and notations. All definitions can be found in standard textbooks.

**Pushforward of measures.**    For a measure $\mu$ on a (measurable) space $X$ and a measurable map, $f : X \to Y$ the pushforward measure $f_*\mu$ is defined by $(f_*\mu)(A) = \mu(f^{-1}(A))$ for any measurable set $A \subset Y$. Here $f^{-1}(A) = \{x \in X \mid f(x) \in A\}$ denotes the preimage of $A$ under $f$. Sometimes the pushforward measure is denoted by $f^\# \mu$. Note that no further restrictions on $f$ are necessary, in particular $f$ does not need to be invertible. One important property that we will use frequently is the relation $(g \circ f)_*\mu = f_*(g_*\mu)$.

Note that if $s \sim \mathbb{P}$, i.e., $s$ has distribution $\mathbb{P}$ then $f(s) \sim f_*\mathbb{P}$. Indeed, this is obvious as $P(f(s) \in A) = P(s \in f^{-1}(A)) = \mathbb{P}(f^{-1}(A))$. In the context of ICA it is convenient to mostly talk about distributions and push-forwards as we typically never observe pairs $(s, f(s))$. For later usage we also recall the transformation formula for random variables. If $f \in C^1(\mathbb{R}^n, \mathbb{R}^n)$ is an invertible diffeomorphism and $\mathbb{Q} = f_*\mathbb{P}$ where $\mathbb{P}$ and $\mathbb{Q}$ have density $p$ and $q$ respectively then

$$q(y) = p(f^{-1}(y))|\operatorname{Det} Df^{-1}(y)|. \tag{11}$$

Note that the potentially more familiar version for random variables reads as follows. Let $X$ and $Y$ be random variables satisfying $Y = f(X)$, then their densities are related by

$$p_Y(y) = p_X(f^{-1}(y))|\operatorname{Det} Df^{-1}(y)|. \tag{12}$$

**Diffeomorphisms.**    Let $U, V \subset \mathbb{R}^d$. A diffeomorphism from $U$ to $V$ is a bijective map $f : U \to V$ such that $f \in C^1(U, V)$ and $f^{-1} \in C^1(V, U)$. Note that it is not sufficient to assume that $f$ is bijective and in $C^1(U, V)$, the classic counterexample being $f(x) = x^3$. A sufficient condition is that $Df(x)$ is an invertible matrix for all $x$. Sometimes we loosely speak about diffeomorphisms $f : U \to \mathbb{R}^d$ which should be understood as $f$ being a diffeomorphism on its image $f(U)$.

**Vector fields and flows**    Vector fields can be introduced nicely in the language of differential geometry. However, we think that for the purpose of this paper it is more appropriate to give a more down to earth discussion focused on $\mathbb{R}^d$. We refer to Lee [2003] for a general introduction. A vector field is a map $X : \mathbb{R}^d \to \mathbb{R}^d$. Our reason to consider vector fields is that they can be used to describe smooth transformations of $\mathbb{R}^d$. We will assume that $X$ is Lipschitz continuous. We define the flow of a vector field as a map $\Phi : \mathbb{R} \times \mathbb{R}^d \to \mathbb{R}^d$ such that

$$\Phi_0(x) = x, \qquad \partial_t \Phi_t(x) = X(\Phi_t(x)). \tag{13}$$

Let us remark concerning the notation that it is convenient to put the $t$ argument below, i.e., we write $\Phi_t(x) = \Phi(t, x)$. Moreover, when applying differential operators $D$ they will by default only act on the spatial variable $x$, i.e., $D\Phi_t(x)$ denotes the derivative of the function $x \to \Phi_t(x)$ for a fixed $t$. Note that the solutions of differential equations can exhibit blow-up so $\Phi_t(x)$ might not be defined for all times $t$. However, when we assume that $X$ is bounded the flow exists globally.

It can be shown that if $X$ is $k$ times differentiable then so is $\Phi_t$ and one can conclude that then $\Phi_t$ is a diffeomorphism ($\Phi$ is bijective by the uniqueness of ordinary differential equations). We will also consider time-dependent vector fields $X : (-T, T) \times \mathbb{R}^d \to \mathbb{R}^d$ where the flows can be defined similarly, replacing $X$ by $X_t$.

We are particularly interested in the action of flows on probability measures, i.e., we consider the measures $(\Phi_t)_*\mathbb{P}$ for some initial measure $\mathbb{P}$. It can be shown that if the density of $\mathbb{P}$ is $p$ then the density $p_t$ of $(\Phi_t)_*\mathbb{P}$ satisfies the continuity equation

$$\partial_t p_t + \operatorname{Div}(p_t X_t) = 0 \quad \text{and} \quad p_0 = p. \tag{14}$$

Here Div denotes the divergence which is defined by $\text{Div } f = \sum_i \partial_i f_i = \text{Tr } Df$. One important consequence is that the flow of the vector field preserves the measure, i.e., $(\Phi_t)_* \mathbb{P}$ iff $\text{Div}(pX_t) = 0$ for all $t$. Moreover, the standard Lebesgue measure in $\mathbb{R}^d$ is preserved if $\text{Div}(X_t) = 0$, i.e., divergence free vector fields generate volume preserving flows and vice versa.

**Additional notation.** We at some places use the notation $[n] = \{1, \ldots, n\}$. We also use the $\mathcal{O}$ notation. Recall that $f(x) = \mathcal{O}(g(x))$ as $x \to \infty$ means that there are constants $x_0$ and $C > 0$ such that $f(x) \leq Cg(x)$ for $x \geq x_0$. Recall that we introduced the notation $C_d = (0, 1)^c$ in the main part and we denoted by $\nu$ the uniform measure on $C_d$. We write iff as a shorthand for 'if and only if'.

# B   MEASURE PRESERVING TRANSFORMATIONS AND SPURIOUS SOLUTIONS

In this section we review the construction of spurious solutions for the ICA problem. We assume that we consider ICA in the class $(\mathcal{F}, \mathcal{P})$ where $\mathcal{F}$ is a function class and $\mathcal{P}$ a class of probability measures. We are interested in understanding the set of spurious solutions for a pair $(f, \mathbb{P})$, i.e., the set of pairs $(f', \mathbb{P}') \in \mathcal{F} \times \mathcal{P}$ such that $f_* \mathbb{P} = f'_* \mathbb{P}'$. We now note that if we can construct $h$ such that $h_* \mathbb{P}' = \mathbb{P}$ and set $f' = f \circ h$ we have $f_* \mathbb{P} = f'_* \mathbb{P}'$. Next we define the subset of right composable functions

$$\mathcal{F}^R = \{f \in \mathcal{F} \mid g \circ f \in \mathcal{F} \text{ for all } g \in \mathcal{F}\} \tag{15}$$

and similarly the subset of left-composable functions

$$\mathcal{F}^L = \{f \in \mathcal{F} \mid f \circ g \in \mathcal{F} \text{ for all } g \in \mathcal{F}\}. \tag{16}$$

We remark that if $\mathcal{F}$ is a group then obviously $\mathcal{F}^L = \mathcal{F}^R = \mathcal{F}$ and the problem reduces to finding measure preserving transformations in $\mathcal{F}$. The classes $\mathcal{F}_{\text{lin}}$, $\mathcal{F}_{\text{nonlinear}}$, $\mathcal{F}_{\text{conf}}$, and $\mathcal{F}_{\text{volume}}$ are all groups. We will comment on $\mathcal{F}_{\text{OCT}}$ below.

We note that if $h \in \mathcal{F}^R$ satisfies $h_* \mathbb{P}' = \mathbb{P}$ then $f' = f \circ h \in \mathcal{F}$ and $f'_* \mathbb{P}' = f_* \mathbb{P}$ so this gives us a spurious solution. A specific case is given by $\mathbb{P} = \mathbb{P}'$ in which case we are looking for measure preserving transformations (MPTs) $h \in \mathcal{F}^R$. Note that such $h$ for a certain $\mathbb{P}$ allows us to construct spurious solutions for all pairs $(f, \mathbb{P})$ with arbitrary $f$.

This implies that if $(\mathcal{F}, \mathcal{P})$ is identifiable with respect to $\mathcal{S}$ then any $h$ as above satisfies $h \in \mathcal{S}$ (if $\mathbb{P}'$ has full support, and otherwise there is a $h' \in \mathcal{S}$ such that $h$ and $h'$ agree in the support of $\mathbb{P}'$).

We consider an example. Let $\mathbb{P}$ be the distribution of the standard Gaussian and $h = R$ for some $R \in O(d)$. Then $h \in \mathcal{F}_{\text{lin}}$ and $h_* \mathbb{P} = \mathbb{P}$ as the standard Gaussian is invariant under rotations. However, such a linear measure preserving transformation does not exist for other $\mathbb{P} \in \mathcal{M}_1(\mathbb{R})^{\otimes d}$. This is the reason that Gaussian distributions are excluded for linear ICA.

Next we observe (as we discussed in the main part) that if the function class $\mathcal{F}$ is stable by right composition with arbitrary component-wise transformation we can turn any admissible latent distribution into another one. This means we can fix a reference measure (we will usually use $\nu$) and then we can find (at least under suitable regularity assumptions) for $\mathbb{P} \in \mathcal{M}_1(\mathbb{R})^{\otimes d}$ maps $h^{\mathbb{P} \to \nu}, h^{\nu \to \mathbb{P}} \in \mathcal{F}^R$ such that $h_*^{\mathbb{P} \to \nu} \mathbb{P} = \nu$, $h_*^{\nu \to \mathbb{P}} \nu = \mathbb{P}$. If we can find an MPT $h \in \mathcal{F}^R$ such that $h_* \nu = \nu$ mixing the coordinates we can then find spurious solutions for any pair $(f, \mathbb{P})$ by considering $f' = f \circ h^{\nu \to \mathbb{P}} \circ h \circ h^{\mathbb{P} \to \nu}$ because then $f'_* \mathbb{P} = f_* \mathbb{P}$ and $f' \in \mathcal{F}$ by definition of $\mathcal{F}^R$.

For the class $\mathcal{F}_{\text{nonlinear}}$ such MPTs exist, we gave one construction in Lemma 1. In Appendix E we will sketch the proof of this result and discuss another construction based on divergence free vector fields. For the class $\mathcal{F}_{\text{volume}}$ we cannot arbitrarily transform the input distribution. However, we can still find coordinate mixing MPTs for every $\mathbb{P}$ (with smooth density). This will be proved in Appendix F. These constructions rule out any (meaningful) identifiability result for $\mathcal{F}_{\text{nonlinear}}$ and $\mathcal{F}_{\text{volume}}$.

There is also a slightly different approach to construct spurious solutions for a pair $(f, \mathbb{P})$. For any MPT $h \in \mathcal{F}^L$ such that $h_* f_* \mathbb{P} = f_* \mathbb{P}$, i.e., transformations that preserve the observational distribution $\mathbb{Q} = f_* \mathbb{P}$, the function $h \circ f$ defines a spurious solution. Note that this gives spurious solutions for all pairs $(f, \mathbb{P})$ such that $f_* \mathbb{P}$ follows the fixed distribution $\mathbb{Q}$. Note that because of their fine-tuning, such spurious solutions can arguably be considered pathological cases instead of key non-identifiability issues. This is inline with how such solutions a considered in the case of linear ICA.

Let us finally have a closer look at $\mathcal{F}_{\text{OCT}}$. It is quite straightforward to see that

$$\mathcal{F}_{\text{OCT}}^L = \mathcal{F}_{\text{conf}}. \tag{17}$$

In particular, all rotations are contained in $\mathcal{F}_{\text{conf}}^L$. More interestingly, we have

$$
\begin{aligned}
\mathcal{F}_{\text{OCT}}^R &= \{f \in \mathcal{F}_{\text{OCT}} \mid Df(x) = P(x)\Lambda(x) \text{ for some } P(x) \in \text{Perm}(d), \Lambda(x) \in \text{Diag}(d)\} \\
&= \{f \in \mathcal{F}_{\text{OCT}} \mid f = P \circ h \text{ where } P \in \text{Perm}_\pm(d) \text{ and } h : \mathbb{R}^d \to \mathbb{R}^d \text{ with} \\
&\qquad h(x) = (h_1(x_1), \ldots, h_d(x_d))^\top \text{ for some } h_i \in C^1(\mathbb{R}, \mathbb{R}) \text{ with } h_i' > 0\}.
\end{aligned}
\tag{18}
$$

The first step can be seen using the chain rule and the definition of $\mathcal{F}_{\text{OCT}}$. The second step follows from the fact that the permutation $P$ is necessarily constant for such a function. This again recovers the fact that OCTs can only be identified up to permutations and coordinate-wise reparametrisations. However, we also conclude that all $h \in \mathcal{F}_{\text{OCT}}^R$ act coordinate-wise (up to a permutation), i.e., do not prevent BSS. This shows that for $\mathcal{F}_{\text{OCT}}$ no completely generic spurious solution based on a single MPT mixing the coordinates exists. This is different from $\mathcal{F}_{\text{nonlinear}}$. We emphasize that this does not rule out the existence of a fine-tuned spurious solution for every (or most) pairs $(f, \mathbb{P})$ because then we only need to find $h$ such that $h_*\mathbb{P} = \mathbb{P}$ and $f \circ h \in \mathcal{F}_{\text{OCT}}$. The relation $f \circ h \in \mathcal{F}_{\text{OCT}}$ of course holds for $h \in \mathcal{F}_{\text{OCT}}^R$ but there will, in general, be many more $h$ for a fixed $f$.

## C  PROOF OF THEOREM 1

We here, for completeness, give a proof of Theorem 1. While this result is well known we think that it makes sense to include a condensed proof because it contains many of the key steps of the more involved proof for conformal maps in the next section and it is not as well known as the proof based on Darmois-Skitovich Theorem which does not generalise to nonlinear functions.

*Proof of Theorem 1.* We assume that $x \overset{\mathcal{D}}{=} As \overset{\mathcal{D}}{=} A's'$. We first assume that the densities $p : \mathbb{R}^n \to \mathbb{R}$ and $q : \mathbb{R}^n \to \mathbb{R}$ of $s$ and $s'$ are $C^2$ functions and $p(x) > 0$ for all $x \in \mathbb{R}^d$. We denote their distributions by $\mathbb{P}$ and $\mathbb{Q}$. By independence of the components we can write $p(x) = \prod_i p_i(x_i)$, $q(x) = \prod_i q_i(x_i)$ for some $C^2$ functions $p_i$ and $q_i$. By assumption we conclude that $((A')^{-1}A)_*\mathbb{P} = \mathbb{Q}$. We denote $B = ((A')^{-1}A)^{-1} = A^{-1}A'$ so that $B_*^{-1}\mathbb{P} = \mathbb{Q}$ and the transformation formula for densities implies that

$$
q(y) = p(By)\,|\text{Det}\,B| \quad \Rightarrow \quad \sum_k \ln(q_k(y_k)) = \sum_k \ln(p_k((By)_k) + \ln|\text{Det}\,B|.
\tag{19}
$$

The main idea of the proof is to use the observation that for $i \neq j$ and all $y$ such that $q(y) \neq 0$

$$
\partial_i \partial_j \ln(q(y)) = \partial_i \partial_j \sum_k \ln(q_k(y_k)) = 0
\tag{20}
$$

i.e., mixed second derivatives of the log density vanish. We plug (19) into this equation and get

$$
\partial_i \ln(q(y)) = \sum_k B_{ki} \ln(p_k)'((By)_k),
\tag{21}
$$

$$
\partial_i \partial_j \ln q(y) = \sum_k B_{kj} B_{ki} \ln(p_k)''((By)_k).
\tag{22}
$$

We now denote $x = By$. Then we get (using that $B$ is invertible) that for all $x$ such that $p(x) \neq 0$

$$
0 = \sum_k B_{kj} B_{ki} \ln(p_k)''(x_k).
\tag{23}
$$

Varying one $x_k$ individually we conclude that each summand is constant which implies that either $B_{kj} B_{ki} = 0$ for all $i \neq j$ or $\ln(p_k)''(x_k)$ is constant. It is straightforward to see that the only probability distribution with $\ln(p)'' = \kappa$ for some constant $\kappa$ are Gaussian distributions. Indeed, note that $\ln(p)'' = \kappa$ for some constant implies that $p(x) = \exp(\kappa x^2/2 + c_1 x + c_2)$. If $p$ is the density of a probability distribution and in particular integrable we see that this implies $\kappa < 0$ and $p$ is a Gaussian density with $\kappa = -\sigma^{-2}$. Note that by assumption at most one component of $\mathbb{P}$ is Gaussian, w.l.o.g., $k = 1$.

As $p_k$ is not a Gaussian for $k > 1$ and thus $\ln(p_k)''$ not constant we conclude

$$
B_{kj} B_{ki} = 0
\tag{24}
$$

for $i \neq j$ and all $k > 1$. Plugging this into (23) we obtain

$$0 = B_{1j} B_{1i} \ln(p_1)''(x_1). \tag{25}$$

Note that if $p_1$ is a Gaussian density then $\ln(p_1)'' = -\sigma^{-2} \neq 0$ so we conclude that in any case $B_{1i} B_{1j} = 0$ for $i \neq j$, i.e., (24) holds as well for $k = 1$.

In other words, at most one entry of every row of $B$ is non-zero. This implies that $B = P\Lambda$ for some $P \in \mathrm{Perm}(d)$ and $\Lambda \in \mathrm{Diag}(d)$. This was to be shown.

Let us clarify what happens when there is more than one Gaussian component. In this case there might be multiple constant non-zero terms in (23) whose contributions can cancel and we cannot conclude that (24) holds for all $k$. This recovers the well known non-uniqueness for Gaussian variables.

It remains to extend the result to distributions whose density is not twice differentiable. By standardizing $s$ and $s'$ we can assume that $B \in \mathrm{O}(n)$. Indeed, when the covariances of $s$ and $s'$ satisfy $\Sigma_s = \Sigma_{s'} = \mathrm{Id}_d$ then $s = Bs'$ implies $B^\top B = B^T \Sigma_{s'} B = \Sigma_s = \mathrm{Id}$. Then $s' \overset{\mathcal{D}}{=} Bs$ implies that for an independent standard normal

$$B(s + N) \overset{\mathcal{D}}{=} s' + BN \overset{\mathcal{D}}{=} s' + N \tag{26}$$

where we used that standard normal variables are invariant under orthogonal maps. Note that $s + N \in \mathcal{P}_{\mathrm{lin}}$. Indeed, $(s_i, N_i) \perp\!\!\!\perp (s_j, N_j)$ implies $(s_i + N_i) \perp\!\!\!\perp (s_j + N_j)$ for $i \neq j$. If $s_i + N_i$ is Gaussian then $s_i$ is Gaussian (consider, e.g., the Fourier transform) so $s + N$ also has at most one Gaussian component. The density of $s + N$ is smooth and pointwise positive, so we can apply the reasoning above to $s + N$ and $s' + N$ and conclude $B \in \mathcal{S}_{\mathrm{lin}}$. $\qquad \square$

# D    PROOFS FOR THE RESULTS ON CONFORMAL MAPS

In this section we give the proofs for Section 3. First, we consider $d > 2$ and then the special case $d = 2$.

## D.1    PROOF OF THEOREM 2

The proof of Theorem 2 uses similar ideas as the proof for Theorem 1, however, the calculations are more involved. The key ingredient is the classification of all conformal maps in $d > 2$ given by Liouville.

**Theorem 6** (Liouville). *If $\Omega \subset \mathbb{R}^n$ is an open connected set and $f : \Omega \to \mathbb{R}^n$ is conformal, then*

$$f(x) = b + \alpha \frac{A(x - a)}{|x - a|^\varepsilon} \tag{27}$$

*where $b, a \in \mathbb{R}^n$, $\alpha \in \mathbb{R}$, $A \in \mathrm{O}(n)$, and $\varepsilon \in \{0, 2\}$.*

Originally this result was shown by Liouville Liouville [1850], a modern treatment is Iwaniec and Martin [2001]. In particular, this shows that conformal maps are (up to translations) rotations or rotations followed by an inversion. in Theorem 6. From there we see that we already dealt with the linear case in Theorem 1, so it is sufficient to focus on the case of nonlinear transformations. In particular, the Theorem will be an easy consequence of the following lemma.

**Lemma 2.** *Suppose $g : \mathbb{R}^d \to \mathbb{R}^d$ is a nonlinear Moebius transformation, i.e., a map as in (27) with $\varepsilon = 2$. Let $s, s'$ be random variables whose distributions are in $\mathcal{P}_{\mathrm{conf}}$. Then $s \overset{\mathcal{D}}{\neq} g(s')$.*

Let us quickly show how it implies Theorem 2 before we prove this lemma.

*Proof of Theorem 2.* We use the same notation as in the proof of Theorem 1. We denote the distribution of $s$ and $s'$ by $\mathbb{P}$ and $\mathbb{Q}$ and we assume $x \overset{\mathcal{D}}{=} f(s) \overset{\mathcal{D}}{=} f'(s')$ with $f, f' : \mathbb{R}^d \to \mathbb{R}^d$ conformal. This implies that $(f'^{-1} f)_* \mathbb{P} = \mathbb{Q}$. We denote $g = ((f')^{-1} f)^{-1} = f^{-1} f'$ so that $\mathbb{P} = g_* \mathbb{Q}$. We apply Liouville's Theorem (recalled in Theorem 6) which implies that

$$g(y) = b + \frac{\alpha A(y - a)}{|y - a|^\varepsilon} \tag{28}$$

where $b, a \in \mathbb{R}^d$, $\alpha \in \mathbb{R}$, $A \in O(d)$, $\varepsilon \in \{0, 2\}$. Using Lemma 2 we conclude that $\varepsilon = 0$ and $g$ is linear. Then we can apply Theorem 1 (using that $\mathcal{P}_{\mathrm{conf}} \subset \mathcal{P}_{\mathrm{lin}}$) and conclude that $\alpha A = P\Lambda$ for a permutation matrix and a diagonal matrix $\Lambda$. Since $g$ is conformal we have that $A$ is orthogonal and all eigenvalues have absolute value 1 which implies that $\Lambda_{ii} = \pm\alpha$ for $1 \leq i \leq d$ and thus $A \in M_{\mathrm{perm},\pm}(\mathbb{R}^{d \times d})$. $\qquad\square$

To prove Lemma 2 we need one technical result that shows that local properties of the density $p_i$ (i.e., properties that hold for $x_i \in O_i$ for some non-empty open sets $O_i$) in fact hold for all $x_i \neq 0$. This will be based on the nonlinearity of the map $g$ combined with the factorisation $p(x) = \prod_i p_i(x_i)$ of the densities. An illustration can be found below in Figure 4.

**Lemma 3.** *Let $O = O_1 \times \ldots \times O_d \subset \mathbb{R}^d$ and $U = U_1 \times \ldots \times U_d \subset \mathbb{R}^d$ where $O_i$ and $U_i$ are non-empty open sets. Let $g(x) = Ax/|x|^2$ for an orthogonal matrix $A$. Assume that $g(O) = U$. Then $U_i$ is either $(0, \infty)$, $(-\infty, 0)$ $(-\infty, 0) \cup (0, \infty)$, or $(-\infty, \infty)$. Moreover, if the $i$-th row of $A$ is not equal to a (signed) standard basis vector then $U_i = \mathbb{R}$.*

Informally the result follows from the fact that coordinate planes $\{x_i = c\}$ are mapped to spheres by $g$ except for $c = 0$. We assume that the boundary of $O$ and $U$ is the union of subsets of hyperplanes. However, $g(O) = U$ then implies that the boundaries of $O$ and $U$ are mapped to each other. Since the boundaries of both sets are the union of subsets of hyperplanes that are mapped to hyperplanes we conclude their boundaries must be a subset of the coordinate axes hyperplanes $H_i = \{x : x_i = 0\}$ (because they would be mapped to spherical caps by $g$). For completeness, we give a careful proof below. Let us highlight that this lemma essentially rules out counterexamples with finite support because we can apply the lemma to the set $\{x \in \mathbb{R}^d : p(x) > 0\}$. This is in contrast to the 2-dimensional case where conformal maps between any two rectangles exist making the proof of the corresponding statement below more difficult.

We now prove Lemma 2.

*Proof of Lemma 2.* The proof is a bit lengthy, so we first give an informal overview of the main steps. As before, we denote the distribution of $s$ and $s'$ by $\mathbb{P}$ and $\mathbb{Q}$. We argue by contradiction, so we assume that $\mathbb{P} = g_*\mathbb{Q}$. By assumption $g(x) = b + \alpha A(x - a)/|x - a|^2$. We now proceed in several steps that constrain the structure of $g$. Let us briefly describe the steps of the proof.

In Step 1 and 2 we eliminate the trivial symmetries of the measure and show that the mild local regularity assumption on the measures imply global regularity.

Then, in Steps 3 and 4 we derive in (38) a condition similar to (23) but more involved.

To exploit this condition we look in Steps 5 and 6 at certain limiting regimes where the terms become much simpler and almost reduce to the condition of the linear case. This allows us to conclude that $A$ is a permutation matrix in Step 7.

Using that $A$ is a permutation matrix in (38) we get in Step 8 a much simpler relation that almost factorizes. This allows us to derive a simple differential equation in Step 9 which restricts the potential densities to a simple parametric form. This allows us to conclude.

**Step 1: Elimination of trivial symmetries.** First we show that we can assume $a = b = 0$ and $\alpha = 1$. We denote the shifts on $\mathbb{R}^d$ by $T_a(x) = x + a$ and the dilations $D_\alpha(x) = \alpha x$. Then we can rewrite $g = T_b \circ D_\alpha \circ g_0 \circ T_{-a}$ with $g_0(x) = Ax/|x|^2$ and therefore $(g_0)_*(T_{-a})_*\mathbb{Q} = (D_{\alpha^{-1}} \circ T_{-b})_*\mathbb{P}$. Since shifts and dilations preserve the class $\mathcal{P}_{\mathrm{conf}}$ it is sufficient to show the result for $g = g_0$. To simplify the notation we drop the 0 in the following and just assume $g(x) = Ax/|x|^2$.

**Step 2: Support and smoothness of distributions.** The goal of this step is to show that under the assumption of Lemma 2 the density of $\mathbb{P}$ and similarly of $\mathbb{Q}$ is positive and $C^2$ away from the coordinate hyperplanes $\{x_i = 0\}$ for a union of quadrants, while it vanishes on the remaining quadrants. This will be a consequence of Lemma 3. Let $U_i = \mathrm{Int}(\mathrm{supp}\mathbb{P}_i)$. By assumption, $\mathbb{P} \in \mathcal{P}_{\mathrm{conf}}$ which entails $U_i \neq \emptyset$. Then $U = U_1 \times \ldots \times U_n = \mathrm{Int}(\mathrm{supp}\,\mathbb{P})$. Define $O_i$ similarly for $\mathbb{Q}$. The relation $g_*\mathbb{Q} = \mathbb{P}$ implies $g(O) = U$. Applying Lemma 3 we conclude that $U$ is the union of quadrants.

The same argument will imply that $p$ is actually $C^2$ away from the coordinate planes. We consider the interior of the set of points where $\mathbb{P}$ has a twice differentiable and positive density and call it $U'$. For $x \in U'$ there is a density $p$ in a neighbourhood of $x$ and it factorizes by the independence assumption. The relation

$$\partial_i^2 p(x)/p(x) = \partial_i^2 p_i(x_i)/p_i(x_i) \tag{29}$$

implies that then $p_i$ is twice differentiable at $x_i$. Vice-versa, if all $p_i$ are twice differentiable at $x_i$ then $p$ is twice differentiable at $x = (x_1, \ldots, x_d)$. This implies that $U' = U_1' \times \ldots \times U_d'$ for some open sets $U_i'$. By definition of $\mathcal{P}_{\mathrm{conf}}$ the sets $U_i'$ are

non-empty. Similarly, we define $O'$. The relation (12) for the densities $p$ and $q$ implies, together with the smoothness of $g$, that $g(O') = U'$. Then we apply again Lemma 3 to conclude that $p$ and $q$ are $C^2$ functions away from the hyperplanes $\{x_i = 0\}$ and if the density is non-zero at a point in a quadrant then it is positive in the complete interior of the quadrant.

Finally, $U_i' = \mathbb{R}$ if the $i$-th row of $A$ has more than one non-zero entry (again by Lemma 3). By definition this means that $p_i \in C^2(\mathbb{R})$ and $p_i(x) > 0$ for such $i$ and all $x$.

Let us emphasize here, that we already finished the proof of Lemma 2 for probability distributions with bounded support. This is more difficult in dimension 2 because there are two-dimensional conformal functions mapping rectangles to rectangles.

A major step in the proof is to show that $A$ is a permutation matrix under the assumptions of the lemma. We define the index set $I \subset [d]$ as the set of all indices $i$ such that the $i$-th row of the matrix $A$ has only one non-zero entry. Our goal is to show that $I = [d]$. We have shown so far that $p_k$ is positive and twice differentiable if $k \notin I$.

The proof in the linear case relied on the relation (23). We now derive a similar relation for non-linear Moebius transformations.

**Step 3: Derivative formulas.** For future reference we note (using $A \in O(n)$) that (for $i \neq j$)

$$(Dg(y))_{kj} = \partial_j g_k(y) = \left( \frac{A}{|y|^2} - 2\frac{Ay \otimes y}{|y|^4} \right)_{kj} = \frac{A_{kj}}{|y|^2} - 2\frac{(Ay)_k y_j}{|y|^4}, \tag{30}$$

$$(\partial_i \partial_j g_k)(y) = -2\frac{A_{kj} y_i + A_{ki} y_j}{|y|^4} + \frac{8(Ay)_k y_i y_j}{|y|^6} \tag{31}$$

$$\mathrm{Det}(Dg(y)) = \mathrm{Det}\,\frac{A}{|y|^2}\,\mathrm{Det}\left( \mathrm{Id} - 2\frac{y \otimes y}{|y|^2} \right) = -|y|^{-2d}. \tag{32}$$

**Step 4: Derivation of a condition for the densities** We apply the same reasoning as in the proof of Theorem 1 to derive partial differential equations for the density $p$. The condition $s = g(s')$, or equivalently $g^{-1}(s) = s'$ and the density relation (12) imply

$$q(y) = p(g(y))|\mathrm{Det}\,\nabla g(y)|. \tag{33}$$

This implies

$$q(y) = p\left(g(y)\right)|\mathrm{Det}\,\nabla(Ay|y|^{-2})| = p\left(g(y)\right)|y|^{-2d} \tag{34}$$

$$\Rightarrow \sum_k \ln(q_k(y_k)) = \sum_k \ln(p_k(g_k(y))) - 2d\ln(|y|). \tag{35}$$

We calculate for $i \neq j$

$$\partial_i \ln(q(y)) = \sum_k (\partial_i g_k)(y)(\ln p_k)'(g(y)) - 2d\partial_i \ln(|y|),$$

$$0 = \partial_j \partial_i \ln(q(y)) = -2d\partial_j \partial_i \ln(|y|) + \sum_k (\partial_i \partial_j g_k)(y)(\ln p_k)'(g_k(y)) \tag{36}$$

$$+ \sum_k (\partial_i g_k)(y)(\partial_j g_k)(y)(\ln p_k)''(g_k(y)).$$

Evaluating the derivatives using (30) and (31) we get

$$0 = 4d\frac{y_i y_j}{|y|^4}$$

$$+ \sum_k \left( \frac{8(Ay)_k y_i y_j}{|y|^6} - 2\frac{A_{kj} y_i + A_{ki} y_j}{|y|^4} \right) \ln(p_k)'(g_k(y)) \tag{37}$$

$$+ \sum_k \left( \frac{A_{kj}}{|y|^2} - 2\frac{(Ay)_k y_j}{|y|^4} \right) \left( \frac{A_{ki}}{|y|^2} - 2\frac{(Ay)_k y_i}{|y|^4} \right) \ln(p_k)''(g_k(y)).$$

Finally we express the variable $y$ through $x = g(y) = Ay|y|^{-2}$. Note that then $|y| = |x|^{-1}$ and $y = A^{-1}x|x|^{-2}$. Plugging this in the last equation we get

$$
\begin{aligned}
0 &= 4d(A^{-1}x)_i(A^{-1}x)_j \\
&\quad + \sum_k \left(8x_k(A^{-1}x)_i(A^{-1}x)_j - 2|x|^2\left(A_{kj}(A^{-1}x)_i + A_{ki}(A^{-1}x)_j\right)\right)\ln(p_k)' \\
&\quad + \sum_k \left(A_{kj}|x|^2 - 2x_k(A^{-1}x)_j\right)\left(A_{ki}|x|^2 - 2x_k(A^{-1}x)_i\right)\ln(p_k)'' \\
&= 4dA_{mi}x_mA_{lj}x_l \\
&\quad + \sum_k \left(8x_kA_{mi}x_mA_{lj}x_l - 2|x|^2\left(A_{kj}A_{mi}x_m + A_{ki}A_{lj}x_l\right)\right)\ln(p_k)' \\
&\quad + \sum_k \left(A_{kj}|x|^2 - 2x_kA_{lj}x_l\right)\left(A_{ki}|x|^2 - 2x_kA_{mi}x_m\right)\ln(p_k)''
\end{aligned}
\tag{38}
$$

where we used $A^{-1} = A^\top$ as $A$ is orthogonal and we used Einstein summation convention to sum over indices that appear twice (we kept the sum over $k$ for better readability). Note that this expression is not homogeneous in $x$.

**Step 5: Simplifications as $x_r \to \infty$.** We fix an index $1 \leq r \leq d$. The strategy is now to send $x_r \to \infty$ while keeping the other coordinates bounded. We can assume by reflecting coordinates that the quadrant $\{x_i > 0, \forall i\}$ is contained in the support of $\mathbb{P}$ and $p$ has a positive $C^2$ density there. We can then rewrite (38)

$$
\begin{aligned}
0 &= \mathcal{O}(x_r^2) + (8x_r^3 A_{ri}A_{rj} - 4x_r^3 A_{rj}A_{ri})\ln(p_r)' + \mathcal{O}(x_r^2 \ln(p_r)') + \mathcal{O}(x_r^3) \\
&\quad + (A_{rj}x_r^2 - 2x_r A_{rj}x_r)(A_{ri}x_r^2 - 2x_r A_{ri}x_r)\ln(p_r)'' + \mathcal{O}(x_r^3 \ln(p_r)'') \\
&\quad + \sum_{k \neq r} A_{kj}A_{ki}x_r^4 \ln(p_k)'' + \mathcal{O}(x_r^3) \\
&= x_r^4 \sum_k A_{kj}A_{ki}\ln(p_k)'' + 4x_r^3 \ln(p_r)' + \mathcal{O}\left(x_r^3(1 + \ln(p_r)'') + x_r^2 \ln(p_r)'\right).
\end{aligned}
\tag{39}
$$

We conclude that

$$
0 = \sum_k A_{kj}A_{ki}\ln(p_k)'' + \frac{4A_{ri}A_{rj}\ln(p_r)'}{x_r} + \mathcal{O}\left(\frac{1 + \ln(p_r)''}{x_r} + \frac{\ln(p_r)'}{x_r^2}\right).
\tag{40}
$$

By varying $x_k \neq x_r$ this almost implies that $A_{kj}A_{ki}\ln(p_k)'' = c_k$ for some constant whenever $p_k > 0$ and twice differentiable. However, for this we need to show that the terms hidden in $\mathcal{O}(\cdot)$ are really negligible, i.e. $\ln(p_r)''$ and $\ln(p_r)'/x_r$ are bounded as $x_r \to \infty$ so that the remainder becomes $o(1)$ which we will establish.

Note that if such a relation could be derived we could conclude, similarly to the linear case, that $A$ is a permutation matrix.

**Step 6: Boundedness of $\ln(p_r)''$ and $\ln(p_r)'/x_r$.** Recall that $I$ is the set of indices such that the $i$-th row of $A$ has only one non-zero entry. Let $r \notin I$ and pick $j, i$ such that $A_{rj}A_{ri} \neq 0$. Fix all coordinates $x_k$ except $x_r$ so that $p_k$ is positive and twice differentiable at $x_k$. Then we can express (40) as

$$
\ln(p_r)'' + \frac{4\ln(p_r)'}{x_r} = R(x_r)
\tag{41}
$$

where the remainder term $R$ contains the remaining terms. The expression $R$ of course depends on the other coordinates $x_k$ for $k \neq r$ but since they are considered fixed here we can view $R$ as a function of $x_r$ alone.

Equation (40) then implies that there is $M > 0$ sufficiently large such that for $x_r > M$ the remainder term $R(x)$ satisfies for some constant $c > 0$.

$$
|R(x)| \leq \frac{1}{2}|\ln(p_r)''| + \frac{|\ln(p_r)'|}{x_r} + c.
\tag{42}
$$

Here the last constant term bounds the $x_r^{-1}$ contribution. Suppose $\ln(p_r)' \leq 0$. Then we can bound

$$
\begin{aligned}
0 &= \ln(p_r)''(x_r) + \frac{4\ln(p_r)'(x_r)}{x_r} - R(x_r) \\
&\leq \ln(p_r)''(x_r) + \frac{1}{2}|\ln(p_r)''(x_r)| + \frac{4\ln(p_r)'(x_r)}{x_r} + \frac{|\ln(p_r)'(x_r)|}{x_r} + c \\
&\leq \ln(p_r)''(x_r) + \frac{1}{2}|\ln(p_r)''(x_r)| + c
\end{aligned}
\tag{43}
$$

We find that $\ln(p_r)'' \geq -2c$ (for $\ln(p_r)'' > 0$ this is clear, and otherwise we can absorb the absolute value part). We conclude by integration that for all $x_r > M$ (note that the bound is trivially true if $\ln(p_r)' > 0$)

$$
\ln(p_r)'(x_r) \geq \min(0, \ln(p_r)'(M)) - 2c(x_r - M).
\tag{44}
$$

Similarly we can bound for $\ln(p_r)'(x_r) \geq 0$

$$
\begin{aligned}
0 &= \ln(p_r)''(x_r) + \frac{4\ln(p_r)'(x_r)}{x_r} - R(x_r) \\
&\geq \ln(p_r)''(x_r) - \frac{1}{2}|\ln(p_r)''(x_r)| + \frac{4\ln(p_r)'(x_r)}{x_r} - \frac{|\ln(p_r)'(x_r)|}{x_r} - c \\
&\geq \ln(p_r)''(x_r) - \frac{1}{2}|\ln(p_r)''(x_r)| - c
\end{aligned}
\tag{45}
$$

implying $\ln(p_r)''(x_r) \leq 2c$ for $x_r \geq M$ such that $\ln(p_r)'(x_r) > 0$. We obtain

$$
\ln(p_r)'(x_r) \leq \max(0, \ln(p_r)'(M)) + 2c(x_r - M).
\tag{46}
$$

Together the last two steps imply that $|\ln(p_r)'(x_r)| \leq C + Cx_r$ for some $C > 0$ and $x_r > M$. Going back to (41) we conclude that there is $C > 0$ such that $|\ln(p_r)''(x_r)| \leq C$ for $x_r > M$. We conclude that for $r \notin I$ and all $i \neq j$

$$
0 = \sum_k A_{kj}A_{ki}\ln(p_k)'' + \frac{4A_{ri}A_{rj}\ln(p_r)'}{x_r} + O\left(x_r^{-1}\right).
\tag{47}
$$

**Step 7: $A$ is a permutation matrix.** If $I = [n]$ we are done. So there is $r \notin I$ and using (47) we conclude by varying $x_k \neq x_r$ that

$$
A_{kj}A_{ki}\ln(p_k)'' = c
\tag{48}
$$

for some constant (depending on $k$, $i$, and $j$). Note that if we assumed that at most one $p_k$ is a Gaussian density we could conclude as in the linear case. However, this assumption is not necessary, as we will now show.

By definition, $A_{kj}A_{ki}\ln(p_k)'' = 0$ for $k \in I$ because there is only one non-zero entry in row $k$ of $A$. We have seen in Step 2 that for $k \notin I$ the density $p_k \in C^2(\mathbb{R})$ and is positive. By assumption, we can find $i \neq j$ such that $A_{kj}A_{ki} \neq 0$ for $k \notin I$. The relation (48) then implies for $k \notin I$ that $\ln(p_k)''(x_k) = \beta_k$ for some constant $\beta_k < 0$ ($p_k$ is a probability density) and all $x_k \in \mathbb{R}$. Then $\ln(p_k)'(x_k) = \beta_k x_k + \gamma_k$ for some constant $\gamma_k$. With $x_r \to \infty$ we conclude from (47) that for $r \notin I$

$$
0 = \sum_{k \notin I} A_{kj}A_{ki}\beta_k + 4A_{ri}A_{rj}\beta_r.
\tag{49}
$$

Summing this over $r \notin I$ we get

$$
0 = \sum_{r \notin I}\left(\sum_{k \notin I} A_{kj}A_{ki}\beta_k + 4A_{ri}A_{rj}\beta_r\right) = (d - |I| + 4)\sum_{k \notin I} A_{kj}A_{ki}\beta_k.
\tag{50}
$$

Dividing (50) by $d - |I| + 4$ and subtracting it from (49) we conclude that

$$
A_{ri}A_{rj}\beta_r = 0
\tag{51}
$$

for all $r \notin I$ and all $i \neq j$. Since $\beta_r$ is non-zero this implies that $A_{ri}A_{rj} = 0$ for $i \neq j$ and thus $r \in I$, a contradiction. This establishes $I = [d]$ and thus $A$ is a signed permutation matrix. By permuting and reflecting the coordinates of $\mathbb{P}$ we can assume $A = \mathrm{Id}$ in the following.

**Step 8: Simplifications in** (38) **for** $A = \mathrm{Id}$**.**    First we remark that for $A = \mathrm{Id}$ the function $g$ leaves the quadrants invariant. It is thus sufficient to consider the case where $p$ and $q$ vanish outside $\{x_i > 0 \; \forall i\}$ and show that no solutions of $s = g(s')$ exist under this condition. Using step 2 we can assume that $p_i(x_i) > 0$ for all $x_i > 0$. In the following all domain are assumed to be the positive half-line. For $A = \mathrm{Id}$ the condition (38) becomes for $x = (x_1, \ldots, x_d)$ such that $x_i > 0$

$$
0 = 4dx_i x_j - 2|x|^2 x_i \ln(p_j)' - 2|x|^2 x_j \ln(p_i)' - 2|x|^2 x_j x_i (\ln(p_i)'' + \ln(p_j)'')
$$
$$
+ \sum_k 8x_k x_i x_j \ln(p_k)' + \sum_k 4x_k^2 x_i x_j \ln(p_k)''. \tag{52}
$$

Dividing this by $2x_i x_j$ we obtain

$$
0 = 2d - |x|^2 \left( \frac{\ln(p_j)'}{x_j} + \ln(p_j)'' + \frac{\ln(p_i)'}{x_i} + \ln(p_i)'' \right) + \sum_k 4x_k \ln(p_k)' + 2x_k^2 \ln(p_k)''. \tag{53}
$$

We now assume that $d > 2$. Let $i$, $j$, and $r$ be pairwise different. Using the last display with $i$, $j$ and $j$, $r$ and subtracting the resulting equations we obtain

$$
0 = |x|^2 \left( \frac{\ln(p_i)'}{x_i} + \ln(p_i)'' - \frac{\ln(p_r)'}{x_r} - \ln(p_r)'' \right). \tag{54}
$$

Varying $x_r$ and $x_i$ independently and since $i \in [n]$ is arbitrary we conclude that there is a constant $\kappa$ such that

$$
\frac{\ln(p_i)'}{x_i} + \ln(p_i)'' = \kappa \tag{55}
$$

for all $i$ and $x_i > 0$.

**Step 9: Conclusion for** $n > 2$**.**    The solutions of the ODE $y(x)/x + y'(x) = \kappa$ are given by

$$
\frac{\alpha}{x} + \frac{\kappa x}{2} \tag{56}
$$

where $\alpha$ is any constant. We conclude that there are constants $\alpha_j$ such that

$$
\ln(p_j)(x) = \alpha_j \ln(x) - \kappa \frac{x^2}{4} + c \Rightarrow p_j(x) \propto x^{\alpha_j} e^{-\frac{\kappa x^2}{4}}. \tag{57}
$$

This implies that

$$
q(y) = p(y/|y|^2)|y|^{-2d} \propto \prod_j \frac{y_j^{\alpha_j}}{|y|^{2\alpha_j}} e^{-\frac{\kappa y_j^2}{4|y|^4}} |y|^{-2d}. \tag{58}
$$

By applying the main argument to $q$ we infer that $q_j$ has to have again the same structure as in (57) so we conclude that $\kappa = 0$ and $\sum_j \alpha_j = -d$. Alternatively, one directly sees that $q$ only factorizes as $q(y) = \prod q_i(y_i)$ if those conditions hold. It is easy to see that those densities satisfy the assumptions. However, $x^\alpha$ is never integrable so there are no probability distributions satisfying the relations $s = g(s')$ This ends the proof for $d > 2$.

**Step 10: Conclusion for** $d = 2$**.**    For $n = 2$ we cannot simplify (53) by considering indices $i \neq j \neq k$. Instead, we directly exploit (53) to obtain a similar conclusion. Similarly to the argument in Step 6 it can be shown that $\ln(p_r)''$ and $\ln(p_r)'/x_r$ are bounded for $x_r$ away from 0. Then we consider $x_1 \to \infty$ in (53) and divide by $x_1^2$. We get (using $\{i, j\} = \{1, 2\}$)

$$
0 = \left( \frac{\ln(p_1)'}{x_1} + \ln(p_1)'' + \frac{\ln(p_2)'}{x_2} + \ln(p_2)'' \right) + 4\frac{\ln(p_1)'}{x_1} + 2\ln(p_1)'' + O(x_1^{-1}). \tag{59}
$$

By varying $x_2$ we conclude just as for $d > 2$ that $\frac{\ln(p_2)'}{x_2} + \ln(p_2)''$ is constant. We conclude as before. $\qquad \square$

Let us now prove the geometric result from Lemma 3 above.

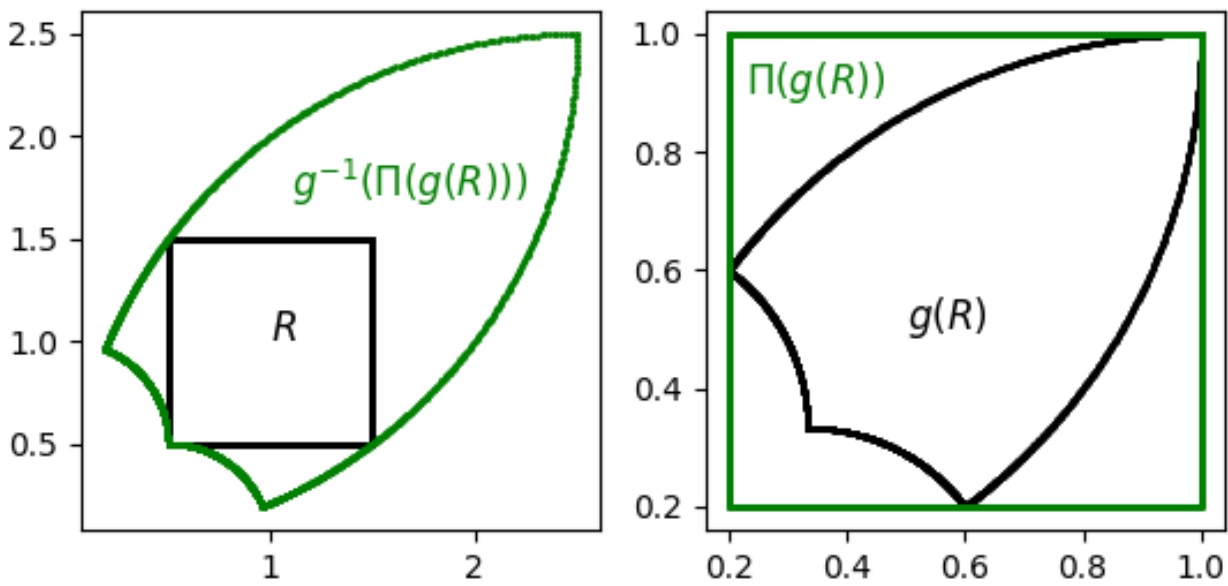

Figure 4: The black rectangle $R$ on the left is mapped by a conformal map to the black shape $g(R)$ on the right. When mapping the smallest rectangle $\Pi(g(R))$ containing $g(R)$ (green rectangle on the right) back to $g^{-1}(\Pi(g(R)))$ (green shape on the left) we obtain a larger set.

*Proof.* The main idea of the proof is that a box contained in $U$ after inversion is distorted so that its convex hull (contained in $O$) is strictly bigger than the box image so that inverting backwards gives us a bigger box in $U$ except for some special cases. An illustration of this argument is shown in Figure 4. The formal argument below is slightly technical. An illustration of the actual argument can be found in Figure 5.

To simplify the notation we write $\iota$ for the inversions $x \rightarrow x/|x|^2$. Then we get $g = A \circ \iota = \iota \circ A$. We consider the projections $\pi_i : \mathbb{R}^d \rightarrow \mathbb{R}$ projecting on the $i$-th coordinate. We consider a map $\Pi$ on subsets of $\mathbb{R}^d$ defined by $\Pi(M) = \pi_1(M) \times \ldots \times \pi_d(M)$. Let $\mathcal{C}$ denote the convex hull of a set. Let $R \subset O$ be a (connected) box. Then $g(R) \subset U$ implies $\Pi(g(R)) \subset \Pi(U) = U$. Since $g(R)$ is connected $\Pi(g(R))$ is convex and thus $\mathcal{C}(g(R)) \subset \Pi(g(R))$. We conclude that $g^{-1}(\mathcal{C}(g(R)) \subset g^{-1}(\Pi(g(R)) \subset g^{-1}(U) \subset O$. As $A$ is linear we have $A^{-1}\mathcal{C}A(M) = \mathcal{C}(M)$ for any set $M \subset \mathbb{R}^d$. Thus, we get $g^{-1}(\mathcal{C}(g(R)) = \iota\mathcal{C}\iota(R)$.

W.l.o.g. we now suppose that there is a box $R = (x_1, y_1) \times \ldots (x_d, y_d) \subset O$ with $0 < x_i < y_i$. We write $x' = (x_2, \ldots, x_d)$, $y' = (y_2, \ldots, y_d)$. We consider the point

$$z = \frac{1}{2}\iota((y_1, x')^\top) + \frac{1}{2}\iota((y_1, y')^\top). \tag{60}$$

Clearly $z \in \mathcal{C}\iota R$. Then

$$\iota(z) = \iota\left(\frac{1}{2}\iota((y_1, x')^\top) + \frac{1}{2}\iota((y_1, y')^\top)\right) \in \iota\mathcal{C}\iota(R) \tag{61}$$

We calculate

$$\iota\left(\frac{1}{2}\iota((y_1, x')^\top) + \frac{1}{2}\iota((y_1, y')^\top))\right) = \iota\left(\frac{1}{2}\frac{(y_1, x')^\top}{y_1^2 + x'^2} + \frac{1}{2}\frac{(y_1, y')^\top}{y_1^2 + y'^2}\right)$$
$$= \frac{\frac{1}{2}\frac{(y_1, x')^\top}{y_1^2 + x'^2} + \frac{1}{2}\frac{(y_1, y')^\top}{y_1^2 + y'^2}}{\left\|\frac{1}{2}\frac{(y_1, x')^\top}{y_1^2 + x'^2} + \frac{1}{2}\frac{(y_1, y')^\top}{y_1^2 + y'^2}\right\|^2} \tag{62}$$

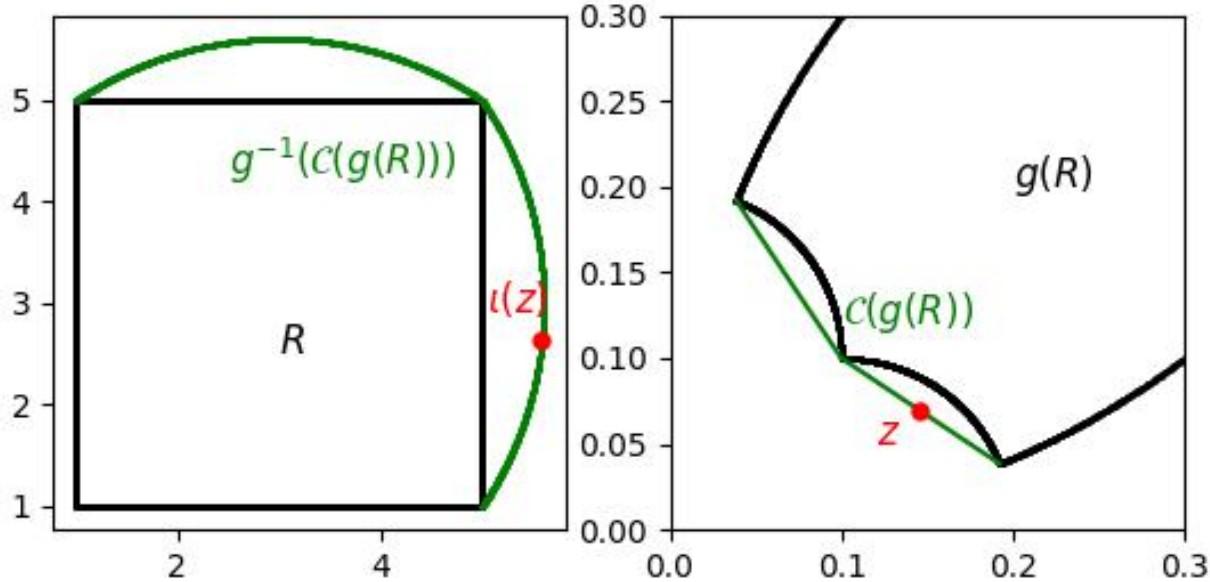

Figure 5: The black rectangle $R$ on the left is mapped by $\iota$ to the black shape $g(R)$ on the right. The green shape on the right shows the convex hull $\mathcal{C}(g(R))$. The point $z$ is defined as in the text and its image $\iota(z)$ lies outside $R$.

We bound (using $x'^2 < y'^2$)

$$
\left\| \frac{1}{2} \frac{(y_1, x')^\top}{y_1^2 + x'^2} + \frac{1}{2} \frac{(y_1, y')^\top}{y_1^2 + y'^2} \right\|^2 < \frac{1}{2} \left\| \frac{(y_1, x')^\top}{y_1^2 + x'^2} \right\|^2 + \frac{1}{2} \left\| \frac{(y_1, y')^\top}{y_1^2 + y'^2} \right\|^2
$$
$$
= \frac{1}{2} \frac{1}{y_1^2 + x'^2} + \frac{1}{2} \frac{1}{y_1^2 + y'^2}.
$$
(63)

Together the last displays imply that

$$
\iota \left( \frac{1}{2} \iota((y_1, x')^\top) + \frac{1}{2} \iota((y_1, y')^\top) \right)_1 > y_1.
$$
(64)

Let $z_1$ be maximal such that $(x_1, z_1) \subset O_1$. Then the reasoning above shows that $z_1 = \infty$. The same reasoning for the other coordinates implies that $R' = (x_1, \infty) \times \ldots \times (x_d, \infty) \subset O$. By applying the same reasoning to sequences of boxes in $g(R)$ approaching the origin we conclude that $U$ is the union of quadrants (and the same holds for $O$).

It remains to prove the last remark. As quadrants are invariant under $\iota$ we have $\iota(O) = O$ and conclude $AO = U$, or equivalently $A^\top U = O$. It is sufficient to show that $0 \in U_i$. For simplicity we assume $R = (0, \infty)^d \subset U$, the generalisation to other quadrants is immediate. Since we assume that the $i$-th row $v_i = A^\top e_i$ of $A$ is not equal to a signed standard basis vector it has at least two non-zero entries. Thus we can find $w$ such that $w \cdot v_i = 0$ and all entries of $w$ are non-zero. Since $w$ is orthogonal to the span of $Ae_i$ there is a vector $\alpha$ such that $A^\top \alpha = w$ and $\alpha_i = 0$. By adding a suitable vector $\beta$ we can ensure that $(\beta + \alpha)_i = 0$, all entries of $A^\top(\alpha + \beta)$ are non-zero and $(\beta + \alpha)_j > 0$ for $j \neq i$. The second condition can be satisfied by picking the entries of $\beta$ one after another. The conditions $(\beta + \alpha)_j > 0$ for $j \neq i$ and $(\beta + \alpha)_i = 0$ imply that $\beta + \alpha \in \overline{U}$ since we assumed $(0, \infty)^d \subset U$. But then $A^\top(\alpha + \beta) \in \overline{O}$ and since $A^\top(\alpha + \beta)$ is strictly contained in a quadrant (all entries are non-zero) we conclude that $A^\top(\alpha + \beta) \in O$ and thus $\alpha + \beta \in U$. This implies $(\alpha + \beta)_i = 0 \in U_i$.

$\square$

# E    PROOFS FOR THE RESULTS ON OCTS

In this section we collect the missing proofs for Section 4.

## E.1 PROOFS OF THEOREMS 3 AND 4

We now consider smooth deformations of a data generating mechanism $x = f(s)$. For this it is helpful to phrase these as flows generated by vector fields. For a brief review of these notions we refer to Appendix A and for an extensive introduction we refer to any textbook on differential geometry. We now give a complete proof of Theorem 3.

*Proof of Theorem 3.* We define $\Psi_t = (\Phi_t)^{-1}$. Then $\Phi_0 = f^{-1}$. We denote the vector field that generates $\Psi_t$ by $X :$ $(0,1)^d \to \mathbb{R}^d$, i.e., $X$ satisfies

$$\partial_t \Psi_t(x) = X_t(\Psi_t(x)). \tag{65}$$

The assumption $(\Phi_t)_* \nu = f_* \nu$ implies $\nu = (\Psi_t)_* f_* \nu$. Then the continuity equation (14) implies that $\text{Div}(X_t) = 0$ on $(0,1)^d$. By assumption, $\Phi_t \in \mathcal{F}_{\text{OCT}}$, which means that

$$(D\Phi_t)^\top D\Phi_t = \Lambda_t \tag{66}$$

where $\Lambda_t : (0,1)^d \to \text{Diag}(d)$ maps to diagonal matrices. By definition of $\Psi$ the flows $\Phi$ and $\Psi$ are related by

$$\Phi(t, \Psi(t,x)) = x \tag{67}$$

for $t \in (-T, T), x \in f(C_d)$. Taking the derivative with respect to $t$ we obtain the relation

$$0 = \partial_t \Phi_t(\Psi(t,x)) + (D\Phi_t)(\Psi(t,x))\partial_t \Psi(t,x) = \partial_t \Phi_t(\Psi(t,x)) + (D\Phi_t)(\Psi(t,x))X(\Psi(t,x)) \tag{68}$$

By setting $s = \Psi(t,x)$ we conclude that

$$\partial_t \Phi_t(s) = -D\Phi_t(s)X_t(s) \tag{69}$$

for $s \in (0,1)^d$. We now want to use this relation in combination with (66). We calculate using (69)

$$\partial_t (D\Phi_t(s))_{ji} = \partial_i(\partial_t \Phi_t(s))_j = -\partial_i \Big( D\Phi_t(s)X_t(s) \Big)_j = -\sum_{k=1}^d \partial_i \left( (D\Phi_t(s))_{jk}(X_t(s))_k \right)$$

$$= -\sum_{k=1}^d (\partial_i \partial_k \Phi_t)_j(s)(X_t(s))_k - \sum_{k=1}^d (D\Phi_t)_{jk}(s))(\partial_i X_t(s))_k \tag{70}$$

$$= -\sum_{k=1}^d (X_t(s))_k \partial_k (D\Phi_t(s))_{ji} - \Big( D\Phi_t(s)DX_t(s) \Big)_{ji}$$

This can be written concisely as

$$\partial_t (D\Phi_t(s)) = -D\Phi_t(s)DX_t(s) - \sum_{k=1}^d (X_t(s))_k \partial_k D\Phi_t(s) \tag{71}$$

This implies

$$\partial_t \Lambda_t = (\partial_t D\Phi_t)^\top D\Phi_t + (\partial_t D\Phi_t)^\top D\Phi_t$$

$$= -(D\Phi_t DX_t)^\top D\Phi_t - (D\Phi_t)^\top D\Phi_t DX_t$$

$$\quad - \sum_{k=1}^d (X_t)_k (\partial_k D\Phi_t)^\top D\Phi_t - \sum_{k=1}^d (X_t)_k (D\Phi_t)^\top (\partial_k D\Phi_t) \tag{72}$$

$$= -(DX_t)^\top \Lambda_t - \Lambda_t DX_t - \sum_{k=1}^d (X_t)_k \partial_k \Lambda_t.$$

Note that the left hand side and the last term are diagonal matrices as $\Lambda_t(s) \in \text{Diag}(d)$ for all $s \in \mathbb{R}^d$ and $t \in (-T, T)$. We conclude that for $i \neq j$ the equation

$$0 = (\Lambda_t DX_t)_{ij} + ((DX_t)^\top \Lambda_t)_{ij} = (\Lambda_t)_{ii}(DX_t)_{ij} + (DX_t)_{ji}(\Lambda_t)_{jj}$$

$$= (\Lambda_t)_{ii} \partial_j (X_t)_i + (\Lambda_t)_{jj} \partial_i (X_t)_j \tag{73}$$

holds. Thus, we obtain a system of first order Partial Differential Equations (PDE) for $X_t$. We can now fix $t$ and drop it from the notation and we write $\Lambda_j = \Lambda_{jj}$. We also fix an $i \in \{1, \dots, d\}$ in the following. Then we can rewrite (73) concisely as

$$\Lambda_i \partial_j X_i + \Lambda_j \partial_i X_j = 0 \quad \text{for } i \neq j. \tag{74}$$

We divide equation (74) by $\Lambda_j$ apply $\partial_j$ and sum over $j \neq i$ to obtain

$$\sum_{j \neq i} \partial_j \left( \frac{\Lambda_i}{\Lambda_j} \partial_j X_i \right) = -\sum_{j \neq i} \partial_j \partial_i X_j = -\partial_i \operatorname{Div} X + \partial_i^2 X_i = \partial_i^2 X_i. \tag{75}$$

This implies that $X_i$ satisfies the wave equation

$$\partial_i^2 X_i - \sum_{j \neq i} \partial_j (a_j \partial_j X_i) = 0 \text{ on } (0, 1)^d \tag{76}$$

where $a_j = \Lambda_i / \Lambda_j$. Note that by assumption $a_j \in C^1((0, 1)^d)$ and $a_j$ is positive because we assumed that $\Phi_t$ are diffeomorphisms implying $\Lambda_j > 0$ (because $D\Phi$ is invertible). Now we use the assumption that $\Phi_t(s) = f(s)$ for $s \notin \Omega \subset (0, 1)^d$. This implies for such $s$ that $s = \Psi_t(\Phi_t(s)) = \Psi_t(f(s))$, i.e., $\Psi_t(f(s))$ is constant for $s \notin \Omega \subset (0, 1)^d$. Using the definition (69) we get that for such $s$ the relation

$$0 = (\partial_t \Psi_t)(\Phi_t(s)) = X(\Psi_t(\Phi_t(s))) = X(s). \tag{77}$$

We conclude $X(s) = 0$ for $s \notin \Omega$.

Now we claim that this together with the PDE (76) implies that $X_i$ vanishes everywhere. We pick $\varepsilon > 0$ such that $\Omega \subset (\varepsilon, 1 - \varepsilon)^d$. Then $X_i$ solves the PDE (76) on $(\varepsilon, 1 - \varepsilon)^d$ with vanishing boundary data and vanishing derivatives at the boundary. Then the uniqueness of solutions for the Cauchy problem for hyperbolic PDE of second order which we stated in Theorem 7 below implies that there is at most one solution. Note that the ellipticity condition in (82) follows by noting that the functions $a_j$ are continuous and positive, and thus $\min_{(\varepsilon, 1-\varepsilon)^d} a_j > 0$.

Since $X_i = 0$ clearly solves the PDE, we conclude that $X_i = 0$. This argument applies to all $t$ and all $i$ so we conclude that $X_t = 0$ for all $t$, i.e., $X_t$ vanishes everywhere. We conclude $\Psi_t = \Psi_0$ and thus $\Phi_t = f$. This ends the proof. $\qquad \square$

For reference, we now state the uniqueness result for second order hyperbolic partial differential equations. Let $U \subset \mathbb{R}^n$ open, bounded and let $U_T = U \times (0, T)$. Consider the boundary problem

$$\partial_t^2 u + Lu = f \text{ in } U_T \tag{78}$$
$$u = 0 \text{ in } \partial U \times [0, T] \tag{79}$$
$$u = g, \partial_t u = h \text{ on } U \times \{0\} \tag{80}$$

where $f : U_T \to \mathbb{R}$ and $g, h : U \to \mathbb{R}$ are given functions which we assume to be $C^1$ and $g = 0$ on $\partial U$. The function $u : U_T \to \mathbb{R}^d$ is the unknown. The operator is assumed to be an elliptic operator given by

$$Lu = -\sum_{i,j=1}^n \partial_i (a^{ij}(x, t) \partial_j u) \tag{81}$$

where we assume $a^{ij} \in C^1(\bar{U}_T)$, $a^{ij} = a^{ji}$, and that there is $\theta > 0$ such that

$$\sum_{i,j=1}^n \xi_i \xi_j a^{ij}(x, t) \geq \theta |\xi|^2 \tag{82}$$

for all $(x, t) \in U_T$ and $\xi \in \mathbb{R}^n$. Then the following result holds.

**Theorem 7** (Theorem 4 in Section 7.2 in Evans [2010]). *Under the assumptions above there is a unique weak solution $u$ of the system* (78) *with boundary values as in* (79) *and* (80).

For our purposes it is not necessary to define weak solution let us just emphasize that any classical solution is a weak solution so this implies uniqueness of classical solutions.

The key obstacle to improve upon this result and to remove the compact support condition on $X$ is that the resulting PDE in equation (76) is well posed for the Cauchy initial value problem but it is not well posed for the Dirichlet problem or for mixed Dirichlet and Neumann boundary data. In particular, solutions are, in general, not unique. Furthermore, there are no general uniqueness results for first order systems as in (73). Note that the existence of a non-trivial divergence free solution $X_0$ of (73) does not imply that a non-constant flow $\Phi_t$ exists because this is not sufficient to define the flow for positive times. We now prove Theorem 4.

*Proof of Theorem 4.* The initial part of the proof proceeds as in the proof of Theorem 3 and we keep using the same notation. We now investigate the boundary conditions for equation 76. Let us define $\Psi'_t$ by $\Psi'_0(s) = s$ and $\partial_t \Psi'_t(s) = X_t(\Psi'_t(s))$ so that $\Psi_t = \Psi'_t \circ \Psi_0 = \Psi'_t \circ f^{-1}$. We infer

$$\nu = (\Psi_t)_*(\Phi_t)_* \nu = (\Psi_t)_* f_* \nu = (\Psi'_t)_* f_*^{-1} f_* \nu = (\Psi'_t)_* \nu. \tag{83}$$

So $\Psi'_t$ preserves $\nu$ and we conclude that $\Psi'_t((0,1)^d) = (0,1)^d$. Let us denote by

$$D_i = \{x \in [0,1]^d \,|\, x_i \in \{0,1\}\} \tag{84}$$

the boundary hyperplanes and write $D = \partial C_d = \partial(0,1)^d = \bigcup_i D_i$. As $\Psi'_t$ maps $(0,1)^d$ bijectively to itself we conclude that

$$(X_t)_i = 0 \quad \text{on } D_i. \tag{85}$$

We now focus on $t = 0$ and use the shorthand $X = X_0$. Then the differential equation (74) implies that

$$\partial_i X_j = \Lambda_i / \Lambda_j \partial_j X_i = 0 \quad \text{on } D_i. \tag{86}$$

We conclude that the function $X_i$ solves the following mixed Dirichlet and Neumann type boundary problem

$$\partial_i^2 X_i - \sum_{j \neq i} \partial_j (a_j \partial_j X_i) = 0 \text{ on } (0,1)^d \tag{87}$$

$$\partial_j X_i = 0 \text{ on } D_j \text{ for } j \neq i \tag{88}$$

$$X_i = 0 \text{ on } D_i. \tag{89}$$

Recall here that $a_j = \Lambda_i / \Lambda_j$. So far we have not used any specific assumption except that $\Phi_t$ is a continuous deformation and $\Phi_0 \in \mathcal{F}_{\text{OCT}}$. So the existence of non-trivial continuous deformations implies that a certain hyperbolic PDE has a non-trivial solution. Unfortunately, this type of boundary value problem for hyperbolic equations is not well posed and has not always a unique solution. We now show that in the specific setting of Theorem 4 uniqueness holds. In this case $f$ is linear and

$$Df = R \operatorname{Diag}(\mu_1, \ldots, \mu_d) \tag{90}$$

so

$$\Lambda = (Df)^\top Df = \operatorname{Diag}(\mu_1^2, \ldots, \mu_d^2). \tag{91}$$

This implies

$$a_j = \Lambda_i / \Lambda_j = \mu_i^2 / \mu_j^2, \tag{92}$$

in particular $a_j$ is constant. So the equation (87) becomes a constant coefficient hyperbolic equation which can be solved explicitly.

We can now use Theorem 1 from Dunninger and Zachmanoglou [1967] (and a simple scaling argument) we conclude that the system (87) has a unique solution which is $X_i = 0$ (actually this result is for $X_i = 0$ on $\partial D$ but the proof is still valid). To give an intuition, we note that separation of variable is possible in this setting and all solutions to the boundary value

problem (87) and (88) (i.e., without the boundary condition (89) for $D_i$) can be expressed as a linear combination of the form

$$X_i(s) = f(s_i) \prod_{j \neq i} \varphi_j(s_j) \tag{93}$$

where $\varphi_j$ are eigenfunctions of the problem $\varphi_j'' = \lambda_j \varphi_j$ on $(0, 1)$ and $\varphi_j'(0) = \varphi_j'(1) = 0$. It is easy to see that those are given by $\cos(\pi m_j t)$ where $m_j \in \mathbb{N}_0$ and then $\varphi_j''(s_j) = \pi^2 m_j^2 \varphi_j$. Solving for $f$ we find from (87) that $f$ satisfies the ode

$$f''(s_i) = \left( \sum_{j \neq i} \pi^2 \frac{\mu_i^2}{\mu_j^2} m_j^2 \right) f(s_i). \tag{94}$$

Using now that $X_i(0) = 0$ (by (89)) we conclude $f(0) = 0$ and therefore

$$f(s_i) = C \sin(\pi \alpha s_i) \tag{95}$$

where $\alpha = \sqrt{\sum_{j \neq i} m_j^2 \mu_i^2 / \mu_j^2}$, or equivalently

$$0 = \alpha^2 \mu_i^{-2} - \sum_{j \neq i} m_j^2 \mu_j^{-2}. \tag{96}$$

Now the condition $X_i(s) = 0$ for all $s \in [0, 1]^d$ with $s_i = 1$ is satisfied if and only if $f(1) = 0$ which holds iff $\alpha \in \mathbb{N}_0$. Note that this argument also implies to solutions that are sums of functions as in (93) by linear independence. Then the assumption that $\mu_i^{-2}$ are linearly independent over $\mathbb{Q}$ implies that $\alpha = 0$ (and $m_j = 0$) which implies $X_i = 0$. Note that this argument only applies at $t = 0$ because it heavily relies on the explicit form of $\Phi_0 = f$. However, we can apply the same reasoning to $\partial_t^k X_t$ inductively and then conclude using the assumption that $\Phi_t$ is analytic in $t$.

The complete argument goes as follows. We take the time derivative of equation (72) and get denoting $\dot{X}_t = \partial_t X_t$ and $\dot{\Lambda}_t = \partial_t \Lambda_t$

$$(DX_t)^\top \dot{\Lambda}_t + \Lambda_t DX_t + (D\dot{X}_t)^\top \Lambda_t + \Lambda_t D\dot{X}_t \in \text{Diag}(d). \tag{97}$$

We have seen that $DX_0 = 0$ so we infer

$$(D\dot{X}_0)^\top \Lambda_0 + \Lambda_0 D\dot{X}_0 \in \text{Diag}(d) \tag{98}$$

and $\text{Div}\, \dot{X}_t = \partial_t \text{Div}\, X_t = 0$. The same arguments as before imply $\dot{X}_0 = 0$ on $(0, 1)^d$. By induction all time derivatives of $X_0$ vanish and using this in (69) we conclude that the Taylor expansion of $\Phi_t(s)$ at $t = 0$ disappears and since we assumed $\Phi_t$ to be analytic in $t$ we conclude. $\qquad \square$

## E.2 PROOFS FOR THE CONSTRUCTION OF SPURIOUS SOLUTIONS

Finally, we show how flows can be used to construct families of solutions to the ICA problem. This section contains the technical results missing in the overview given in Appendix B.

The first construction was described in Lemma 1. Let us for completeness give a proof (we emphasize again that this result is essentially taken from Hyvärinen and Pajunen [1999]).

*Sketch of proof of Lemma 1.* Note that it is sufficient to show that the maps $h_{R,a}$ are volume preserving for fixed $t$ so we ignore the time argument. It is easy to see that $h_{R,a}$ is bijective (the inverse is given $h_{Q,a}$ where $Q(t, r) = R(t, r)^{-1}$). Then we only need to show that $\text{Det}\, Dh_{R,a}(s) = 1$ for all $s$. We calculate (denoting $r = |s - a|$)

$$
\begin{aligned}
(Dh_{R,a}(s))_{ij} = \partial_j (h_{R,a})_i &= R(r)_{ij} + \sum_k (\partial_j R)_{ik}(|s - a|)(s - a)_k \\
&= \partial_j (h_{R,a})_i = R(r)_{ij} + \sum_k (\partial_r R)_{ik}(r)(s - a)_k \partial_j |s - a|.
\end{aligned} \tag{99}
$$

We conclude (writing $R' = \partial_r R$)

$$Dh_{R,a}(s) = R(r) + R'(r)(s-a) \otimes \nabla|s-a| = R(r) + \frac{1}{|s-a|}R'(r)(s-a) \otimes (s-a). \tag{100}$$

Then we obtain, using the matrix determinant lemma for rank 1 updates ($\mathrm{Det}(A + u \otimes v) = (1 + u \cdot A^{-1}v)\,\mathrm{Det}\,A$

$$\mathrm{Det}\,Dh_{R,a}(s) = \left(1 + \frac{1}{|s-a|}(s-a)R(r)^\top R'(r)(s-a)\right)\mathrm{Det}(R(r)). \tag{101}$$

Now we use that $R(r) \in \mathrm{O}(d)$ so $\mathrm{Det}(R(r)) = 1$ and $R(r)^{-1} = R(r)^\top$. Differentiating $R(r)^\top R(r) = \mathrm{Id}_d$ with respect to $r$ we conclude that $R(r)^\top R'(r)$ is skew which implies

$$(s-a)R(r)^\top R'(r)(s-a) = 0. \tag{102}$$

We have therefore shown $\mathrm{Det}\,Dh_{R,a}(s) = 1$, completing the proof. $\qquad\square$

Now we give another construction that also establishes Fact 1 based on suitable divergence free vector fields. All we need to construct is divergence free vector fields with compact support. Consider any smooth function $\varphi : \mathbb{R}^d \to \mathbb{R}$ such that its support is contained in $\Omega$. Then we consider the vector fields $X^{ij} : \mathbb{R}^d \to \mathbb{R}^d$ for $1 \le i < j \le d$ given by

$$X_i^{ij} = \partial_j\varphi, \quad X_j^{ij} = -\partial_j\varphi, \quad X_k^{ij} = 0 \quad \text{for } k \notin \{i,j\}. \tag{103}$$

Then we get $\mathrm{Div}\,X^{ij} = \partial_i\partial_j\varphi - \partial_j\partial_i\varphi = 0$. So those vector fields are divergence free and we conclude that the space

$$\mathcal{X} = \{X : \mathbb{R}^d \to \mathbb{R}^d | \mathrm{supp}(X) \subset \Omega,\ \mathrm{Div}\,X = 0\} \tag{104}$$

is infinite dimensional. Every $X \in \mathcal{X}$ generates a flow $\Phi_t$ defined by

$$\partial_t\Phi_t = X(\Phi_t), \quad \Phi_0(s) = s. \tag{105}$$

Using equation (14) we conclude that $(\Phi_t)_*\nu = \nu$ because $\nu$ has a constant density and the support condition of $X$ ensures that $\Phi_t((0,1)^d) = (0,1)^d$. Then the family $f_t = f \circ \Phi_t$ has the property that $(f_t)_*\nu = f_*\nu$. Note that this construction can be easily generalised to source distributions $\mathbb{P}$ with differentiable density $p$. In this case the condition $\Phi_t\mathbb{P} = \mathbb{P}$ is satisfied when $\mathrm{Div}(pX) = 0$. Clearly it is sufficient to consider $X = Y/p$ where $Y \in \mathcal{X}$ (assuming that $p > c$ for some $c > 0$ on $\Omega$).

# F  PROOFS FOR THE RESULT ON VOLUME PRESERVING MAPS

Next we show that this construction can be generalised to volume preserving transformations and we prove Theorem 5. Note that in the special case that the distribution of $s$ is $\nu$ the construction above already works. This is a special case because the condition $(f_t)_*\nu = f_*\nu$ already implies that $f_t$ is volume preserving as soon as $f$ is volume preserving as the density of $\nu$ is constant. So in this case the condition that $f_t$ is volume preserving and $(f_t)_*\nu = f_*\nu$ essentially agree which is not the case for general base measures.

*Proof of Theorem 5.* We define a suitable vector field explicitly. Consider $X^{ij} : \mathbb{R}^d \to \mathbb{R}^d$ for $1 \le i < j \le d$ defined by

$$X_k^{ij} = \begin{cases} \partial_j p & k = i \\ -\partial_i p & k = j \\ 0 & k \notin \{i,j\}. \end{cases} \tag{106}$$

An illustration of this vector field is given in Figure 6. We consider the family of functions $f_t = f \circ \Phi_t^{ij}$ where the flow $\Phi_t^{ij}$ is defined by $\Phi_0^{ij}(s) = s$ and $\partial_t\Phi_t^{ij}(s) = X(\Phi_t^{ij}(s))$. Note that boundedness of $\nabla p$ and $p \in C^2$ imply that $\Phi_t$ exists globally and defines a diffeomorphism. We claim that $\Phi_t^{ij}$ satisfies $\mathrm{Det}\,\Phi_t^{ij}(s) = 1$ for all $s$ and $(\Phi_t^{ij})_*\mathbb{P} = \mathbb{P}$. The former condition means that $\Phi^{ij}$ preserves the standard volume (Lebesgue-measure) which is the case if $\mathrm{Div}(X^{ij}) = 0$ while the second relation is satisfied if $\mathrm{Div}(pX^{ij}) = 0$ by equation (14). We calculate

$$\mathrm{Div}\,X^{ij} = \partial_i\partial_j p - \partial_j\partial_i p = 0. \tag{107}$$

We also find

$$\mathrm{Div}(X^{ij}p) = p\,\mathrm{Div}(X^{ij}) + X^{ij} \cdot \nabla p = \partial_j p\partial_i p - \partial_i p\partial_j p = 0. \tag{108}$$

This ends the proof. $\qquad\square$

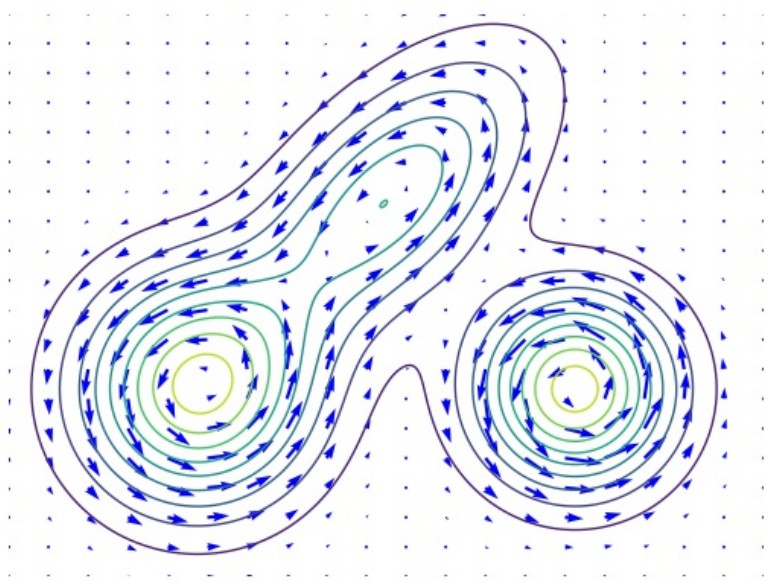

Figure 6: A sketch of the vector fields $X^{ij}$ for $d = 2$ constructed in the proof of Theorem 5. The closed lines are level lines of the probability density (which is a Gaussian mixture here). Note that the vector field is parallel to the level lines and its magnitude proportional to the norm of the gradient of the density.

To give an example, we consider $d = 2$ and $\mathbb{P}$ with rotation invariant density $p(s) = p(|s|)$. Then $X(s) = f(|s|)s^\perp$ where $s^\perp = (s_2, -s_1)^\top$ and the flow lines are circles around the origin where the speed depends on the radius through the derivative of $p(|s|)$. Let us add some remarks concerning this result.

*Remark* 1. 1. The constructed flows are non-trivial, i.e., not constant because the probability density cannot be constant (as we assumed it to be $C^2$) and thus $X^{ij}$ is not identically vanishing.

2. It is easy to see (e.g., through the example above) that the flows $\Phi^{ij}$ will, in general, mix the coordinates $i$ and $j$ thus this really shows that ICA is not identifiable for volume preserving maps.

3. While we construct a finite family of solutions they can be combined, e.g.,

$$f' = f \circ \Phi_{t_1}^{i_1 j_1} \circ \ldots \cdot \Phi_{t_k}^{i_k j_k} \tag{109}$$

to yield a large space of solutions.

4. By choosing coordinates cleverly, it is possible to construct a vector field $X$ satisfying $\mathrm{Div}(X) = \mathrm{Div}(pX) = 0$ with compact support. So even knowing $f$ close to the boundary of the support of $\mathbb{P}$ is not sufficient to uniquely identify $f$.

5. While it is not possible to identify ICA using volume preserving transformations, it can be possible to identify $f(s)$ for certain values of $f$ if $\mathbb{P}$ is known. If $p$ has a unique maximum at $s_0$ then $x_0 = f(s_0)$ will be the point with the largest density of $x$ because volume preserving transformations transform the density trivially (see (12)).