# OpenReview forum: "Function Classes for Identifiable Nonlinear Independent Component Analysis"
_auai.org/UAI/2022/Workshop/CRL — CRL@UAI 2022 Poster_

### Official Review · Reviewer_6CAf · 2022-06-27
**General comments**

**Rating:** 8
**Confidence:** 2

**Review:**

## Summary

The main contributions of the paper are new theoretical identifiability proofs for nonlinear ICA without the use of auxiliary variables when the mixing function class is restricted to particular types. The contributions are stated below:

- definition of the concept of local identifiability;
- proof that orthogonal maps are locally identifiable;
- proof that volume-preserving transformations are not locally identifiable;
- proof that conformal maps are identifiable for d>2.

## Pros

- The paper is well-written and presents visualizations of the function classes studied.
- The contribution is original and could have a high impact on the community.
- The statements, lemmas, and proofs are written rigorously.
- It is useful and original to learn that general volume-preserving transformations are not locally identifiable.

## Cons

- The main contributions of the paper are the proofs, which are given in the appendix. It is difficult for the reader to recognize and verify the added value of this work from the main paper solely. Brief argumentations or summaries of at least the most important proofs could improve the quality of the paper.
- While the theoretical motivation of the paper is interesting and potentially impactful, it would be interesting to discuss the practical applications of such function classes. Empirical experimentation of the ideas would substantially make the paper more appealing, but a simple discussion of potential applications and limitations would improve the reach of this work.

## Comment

- The fourth paragraph of the Conclusion has been repeated (from the second paragraph).

---

### Official Review · Reviewer_hKfW · 2022-06-28
**Identifiability of conformal maps**

**Rating:** 5
**Confidence:** 4

**Review:**

This paper mainly provides identifiability results for conformal maps when the dimension of sources d>=3, which is an extension of the previous result in cases where d<3 [Hyvarinen and Pajunen, 1999]. Besides, partial results for a more general form, i.e., orthogonal coordinate maps, are presented. The idea is clear, but it would be better if experiments and discussions of the conditions could be conducted, since the practical significance of the constraint of the specific functional form is not obvious. Detailed comments are as follows:

1. When d>=3, conformal maps could be exactly formulated as a specific functional form as shown in Theorem 6 and [Liouville, 1850], i.e., Moebius transformations. Although it is of independent interest to extend the previous identifiability for 2D conformal maps, the practical significance of the proposed identifiability result of this specific functional form might not be obvious. At the same time, it is not clear that conformal maps are of a wider range/more practical than the post-nonlinear case, of which the identifiability result has been proved in [Taleb and Jutten, 1999]. A discussion of the specific constraint of this functional form in practice, or if possible, a real-world analysis of it, could be helpful. I would suggest a similar discussion like that in [Gresele et al. 2021]. But compared to their assumption of independent mechanism analysis, the constraint of conformal maps requires the norms of the columns of the Jacobian of the true mixing function to be exactly the same, which seems to be impossible in practice.

2. In the current manuscript, there are no experimental results, such as simulations or real-world analysis. A direct way to show the practical significance of the constraint is by conducting some real-world experiments. However, this may be hard and the result may be negative because of the reasons mentioned in point 1. In this case, it may be better to conduct some simulations to support the proposed identifiability theory empirically.

3. Recently, [Zheng et al., 2022] proved the identifiability of nonlinear ICA in the completely unsupervised case that there is no auxiliary variable. Their assumption of independent influences seems to be closely-related to orthogonal coordinate transformations, which is more general than conformal maps. Perhaps a brief discussion of this result as a related work could be beneficial for readers to better understand the problem.

4. In the Discussion section, the fourth paragraph is verbatim the same as the second paragraph.


Minors:

1. Several references are broken. For example, 'a map as in (??)' in Lemma 2 and several other '(??)' across the manuscripts.

2. It would be better if the resolution of the figures could be higher.


References:

Hyvärinen, Aapo, and Petteri Pajunen. "Nonlinear independent component analysis: Existence and uniqueness results." Neural networks 12.3 (1999): 429-439.

Liouville, Joseph. "Extension au cas des trois dimensions de la question du tracé géographique." Note VI (1850): 609-617.

Taleb, Anisse, and Christian Jutten. "Source separation in post-nonlinear mixtures." IEEE Transactions on signal Processing 47.10 (1999): 2807-2820.

Luigi Gresele, Julius von Kügelgen, Vincent Stimper, Bernhard Schölkopf, and Michel Besserve. "Independent mechanism analysis, a new concept?" NeurIPS. 2021.

Zheng, Yujia, Ignavier Ng, and Kun Zhang. "On the Identifiability of Nonlinear ICA with Unconditional Priors." ICLR2022 Workshop on the Elements of Reasoning: Objects, Structure and Causality. 2022.

---

### Meta-Review · Program_Chairs · 2022-07-06

**Recommendation:** Accept (Poster)
**Confidence:** 4

**Metareview:**

There is some disagreement between the reviewers: while one believes the work is original and may have a high impact on the community, the other reviewer had some concerns, including the lack of experimental results, and asked for more discussion of related work. Overall, we believe that the work contributes theoretical results which may be of interest and worth presenting and discussing in the context of the workshop. The authors are encouraged to fix the typos and address the comments by the reviewers, and to comment on the potential practical significance of their results.

---

### Decision · Program_Chairs · 2022-07-06

Accept (Poster)